



# An assessment of GIA solutions based on high-precision GNSS velocity field for Antarctica

Wenhao Li[1], Fei Li[1,2], Shengkai Zhang[1], Jintao Lei[3], Qingchuan Zhang [1], Feng Xiao[1], Lexian Yuan[4]

[1]Chinese Antarctic Center of Surveying and Mapping, Wuhan University, Wuhan 430079, China;

[2]State Key Laboratory of Information Engineering in Surveying,Mapping and Remote Sensing,Wuhan University,Wuhan 430079, China;

[3]Department of Land Surveying and Geo-Informatics, Hong Kong Polytechnic University, Hung Hom 999077, Hong Kong, China;

[4]Changjiang Spatial Information Technology Engineering Co., Ltd., Wuhan 430010, China;

*Correspondence to*: Fei Li(fli@whu.edu.cn)

**Abstract.** Past mass loads, especially  LGM(Last Glacial Maximum), may cause the viscoelastic response of the Earth, this phenomenon is the so-called glacial isostatic adjustment (GIA). GIA not only includes the horizontal and vertical motions of the crust but also the shape, the gravity field and rotation axis of the earth. Due to the uncertainties in the ice loading history and the mantle viscosity , modeling GIA will be difficult and challenging in Antarctica. The GPS velocity field provides an
15 effective method to constrain the GIA vertical velocity; however, to obtain the high-precision GPS velocity field, we must consider the effects of common mode error(CME)and the choice of optimal noise model (ONM). We used independent component analysis (ICA) to remove the CME recorded at 79 GPS stations in Antarctica and determined the ONM of GPS time series based on the Akaike information criterion (AIC). Then, the high-precision GPS velocity field is obtained; we used the high-precision GPS velocity field to assess the application of GIA models in Antarctica. The results show that the
20 maximal GPS velocity variation is up to 1.15 mm yr$^{-1}$, and the mean variation is 0.18 mm yr$^{-1}$. We find systematic underestimations of all GIA model velocities in the Amundsen Sea area (ASE). In the north Antarctic Peninsula (NAP), the vertical velocities are underestimated by 6 GIA models but not the WANG model. Because the upper mantle viscosities in the NAP are lower than those in the south Antarctic Peninsula (SAP), the  GPS vertical velocities in NAP regions are larger than SAP regions. In the Filscher-Ronne Ice Shelves(FRIS), the observed GPS velocity and predicted GIA model velocity
are consistent. In East Antarctica (EA), the vertical motion is nonsignificant, and the GIA and ice loading have a small impact in this area.

## 1 Introduction

GIA is the solid Earth's slower viscoelastic response to past changes in ice-ocean loading. GIA influences plate tectonics, crustal displacements, the geoid and regional sea level patterns(Wang et al., 2008;Ivins et al., 2013;Argus et al., 2010;
hao et al., 2016); we can obtain the GIA vertical velocity through forward modeled (Peltier, 2004), inversely modeled (Riva et al. 2009) and observed geodetically (such as GPS, King et al., 2010). In the forward models, the ice model and the





earth model are combined to compute the GIA velocities (Velicogna and Wahr 2006; Sasgen et al., 2007;) and the GIA vertical velocities can also be obtained by inversing other geodetic method, such as satellite altimetry and gravimetry technologies (Riva et al., 2009; Gunter et al., 2014). Differences in predictions of GIA for Antarctica persist due to the uncertainties of forward models in both the deglaciation history and Earth's rheology, but without adequate and accurate

deglaciation history data, Earth structure models are greatly simplified in forward models, and the constraint data are poor in inverse models; thus, large differences in GIA persist for Antarctica. The GPS can record vertical land motion(VLM) and which has been used widely to constrain GIA uplift (Argus et al., 2014a; Peltier et al., 2015) or using a data-driven approach to directly solve for GIA (Wu et al., 2010). The actual GPS velocities are usually affected by two factors: CME and the ONM, considering that the GIA magnitude is small and sensitively to various errors; therefore, when using the GPS velocity

field to assess or extract the GIA signal, we must filter the CME and confirm the ONM. CME are thought to be related to the spatiotemporal distribution containing unmodeled signals and errors, including environmental loading effects (Atmospheric, non-tide, hydrology, etc.) and systematic errors (Dong et al., 2006). The detrimental effects of these errors could be efficaciously reduced after applying filtering.

Wdowinski et al. (1997) introduced Stacking to remove the CME of GPS time series in southern California. However, on

a spatial scale, we cannot describe the physical mechanism and effect of CME quantitatively. Dong et al. (2006)used principal component analysis (PCA) to analyze 5-year GPS time series in southern California. Since then, many researchers used widely PCA and modified PCA to remove the CME of GPS time series (Serpelloni et al., 2013; Shen et al., 2014; He et al., 2015; Li et al., 2015). However, CME derived PCA methods are usually considered to contain colored noise (Dong et al., 2006; Yuan et al., 2008), CME and colored noise dissatisfy Gaussian characteristic. In addition, PCA method is based on

second-order statistics and cannot take full advantage of higher-order statistics. Therefore, PCA filtering would result in contamination when applied to non-Gaussian GPS time series. Relative to PCA, independent component analysis (ICA) can take full advantage of higher-order statistics to exploit the non-Gaussian features of the GPS time series (Hyvärinen & Oja 2000). Ming et al. (2017) adopted ICA for an investigation of 259 GPS stations in China. Liu et al.(2018) used ICA methods to GPS time series and explore the effects of CME in Antarctica. Li et al.(2019) compared the filtering results of

Antarctica GPS residual time series derived from PCA and ICA. Considering the shortcomings of stacking and PCA filters, we apply ICA method to extract the CME of GNSS time series from Antarctica.

The noise model is another important factor which can affect the precision of velocity estimate. Zhang J et al.,1997 and Mao A et al.,1999 shown that the GPS time series not only contain white noise (WN) but also colored noise, e.g., flicker noise (FN) and random walk noise (RW). If we ignored the effects of colored noise, the uncertainty of GPS velocity will be

overestimated by a factor of 4 or even one order of magnitude higher than the signal amplitude (Yuan et al., 2008). For regions with a vast spatial area and complex terrain such as Antarctica, with only one noise model to model all GPS station time series is not reasonably and effectively. In this paper, we adopted five noise models to confirm the ONM for the GPS time series in Antarctica: white noise plus power low noise (WN+PN), white noise plus random walk noise (WN+RW),



white noise plus flicker noise (WN+FN), white noise plus power low noise plus random walk noise (WN+FN+RW), and white noise plus random walk noise plus generalized Gauss-Markov (WN+RW+GGM).

After regional filtering and confirming the ONM, we obtain the high-precision GPS velocity field, and 7 GIA models are assessed by the GPS velocity field: ICE-6G (VM5a) (Argus et al.,2014;Peltier et al.,2015),ICE-5G (VM2_L90) (Peltier et al.,2004:Argus et al.,2010),WANG (Wang et al.,2008),W12a (Whitehouse et al.,2012a、2012b),Geruo13 (Geruo et al.,2013),IJ05-R2 (Ivins et al.,2013)and Paulson07(Paulson et al.,2007),The Geruo13 model has three submodels based on different truncation orders and Gauss filtering radii: (a) truncated to 100 order and no Gauss filtering; (b) truncated to 60 order and 200 km Gauss filtering; and (c) truncated to 40 order and 500 km Gauss filtering. The IJ05-R2 model has two submodels based on different parameters of the Earth model: (a) the lithosphere thickness is 65 km and the viscosity of the lower mantle is $1.5\times 1021$ Pa.s; and (b) the lithosphere thickness is 115 km and the viscosity of the lower mantle is $4\times 1021$ Pa.s. In this paper, we use Geruo13 (100 order) and IJR2-05 (65 km)

The remainder of this paper is organized as follows. In section 2, the data processing and methods are briefly reviewed. The results of the processed GPS data and GIA model assessment are discussed in section 3. In section 4, we discuss the assessment results of different regions. The conclusions of our findings are presented in section 5.

## 2 Data processing and methods

### 2.1 GPS data

The GPS time series are downloaded from the Nevada Geodetic Laboratory http://geodesy.unr.edu/NGLStationPages/GlobalStationList. Based on the distribution and integrity of the GPS time series, we selected 79 GPS stations with a time span from 8 February 2010 to 23 June 2018. The average proportion of missing data of our time series is 25.54%. Figure 1 shows the locations of the 79 GNSS stations in Antarctica. We used third quartile criterion to removed abnormal data from the raw time series and estimated a constant offset and trend in addition to the annual and semiannual terms. We subtracted these terms from the coordinate time series to form the residual time series. For the missing values, we used the regularized expectation-maximization (RegEM) (Schneider 2001) algorithm to interpolate data and obtain the completed time series. We used the completed time series to performed an ICA regional filter. Then, we confirmed the ONM for all GPS time series based on AIC. Finally, we used the high-precision GPS velocity field to assess the 7 GIA models.

### 2.2 ICA filter

As presented by previous authors (Hyvärinen & Oja 2000, Ming et al, 2017), if we want to get statistically independent components (ICs) from mixed-signals, we need to maximize the non-Gaussian characteristic of the output. Each observation $X_i(t) = [x_1(t), x_2(t), \cdots, x_n(t)]^T$ can be considered as a compound of the original signals $S_i(t)$ ,but the weights are different from each other. ICA method would get a separating matrix B, and then the signals $Y_i(t)$ and best estimates of $S_i(t)$. When





applied ICA to GPS time series, each row vector x in X is the GPS coordinate series with trend and mean items removed. To remove CME using ICA, we first need to whiten the GPS time series using Z = MX and E(ZZ$_T$) = I (unit matrix), where M represents the whitened matrix and Z presents the whitened variables, and then we use ICA method to obtain a rotation matrix C, and maximize the non-Gaussian character of the projection Y = C$^T$Z. In this paper, we used the FastICA algorithm

(Hyv ̈arinen 1999; Hyv ̈arinen & Oja 2000) to estimate the IC Y. The detailed description of the ICA filtering can be found in Liu et al.2018 and li et al. 2019.

First, we used a parallel analysis (PA) to confirm how many ICs are statistically significant. The PA analysis is a Monte Carlo-based simulation method which compares the observed eigenvalues with those simulated datasets. If the associated eigenvalue is larger than 99% of the distribution of eigenvalues derived from random data and the IC is retained (Peres-Neto

and Jackson et al.,2005). The PA test results show that the first 8 ICs are statistically significant, so we used the first 8 ICs to perform the ICA filter.

Figure 2 shows the spatial responses of IC1-IC8, from which we can conclude that IC2 has a uniform spatial coherence; IC4 and IC8 are neither completely random nor identical, but they exhibit obvious spatially uniform localized patterns or strong spatial coherence across the network; IC7 exhibits spatially uniform localized patterns in some areas, but the pattern is

15 not entirely uniform, which we suppose is because the unmodeled signals, local effects, and other factors are not considered herein. Based on the spatial response, we used IC2, IC4, IC7, and IC8 to extract CME.

Figure 3 is the residual time series (left) and raw time series (right) of GMEZ before and after applying ICA filtering. Clearly, the scattering in the filtered time series is effectively reduced by the ICA filter, as the mean root mean square (RMS) values decrease from 6.41 mm to 4.46 mm, the maximum reduction in RMS value is 48.41%, the minimum value is 10.83%,

and the mean value is 30.81%.

Figure 4 shows the RMS values of the residual time series before and after applying the ICA filters. The color bar is the the RMS reduction percentage; notably, the RMS values have a larger reduction in the SAP and the FRIS; the reductions in RMS values near the coast are smaller than those in the Antarctic interior regions.

**2.3 AIC criterion and noise analysis**

For the precision of GNSS coordinate time series, the noise model is one of the most important factors, the ONM will be quite different because of local effects among the stations in a network. It is not sufficient to reasonably and effectively model all GNSS station time series with only one noise model. We use the AIC (Bos et al., 2013) to confirm the qualities of the selected noise models. The definition of the log-likelihood is as follows:

$$ln(L) = -\frac{1}{2}[Nln\,(2\pi)+lndet(C)+ \mathrm{r^T}C^{-1}] \tag{1}$$

where $N$ is the actual number of GPS observations (gaps do not contain), and r is the residual vector of the time series. The

30 covariance matrix $C$ is decomposed as follows:

$$C =\sigma^2\bar{C}, \tag{2}$$



where $\bar{C}$ represents the sum of different noise models, and σ is the standard deviation of the conducting WN process, where σ is estimated from the residuals:

$$\sigma = \sqrt{\frac{r^T \bar{C}^{-1} r}{N}} \tag{3}$$

Then, the $AIC$ can be defined as follows:

$$AIC = 2k + 2\,ln(L) \tag{4}$$

Because $detcA = c^N \, detA$, the following formulation is implemented for the likelihood:

$$ln(L) = -\frac{1}{2}[Nln(2\pi) + lndet(\bar{C}) + 2Nln(\sigma) + N]. \tag{5}$$

$k$ is the sum of the parameters in the design matrix and the noise models. The minimum AIC value is the better model.

To determine the ONM for Antarctica, we use a combination of 5 noise models supplied by Hector (Bos et al., 2013) to analyze the 79 GNSS station time series based on AIC: WN+PN, WN+RW, WN+FN, WN+FN+RW, and WN+RW+GGM. The noise analysis results for the corresponding velocities listed in Table 1 show that the WN+FN ONM accounts for 22% (18 GPS stations), the WN+RW+GGM model accounts for 5.1% (4 GPS stations), and the WN+PN model accounts for 72.2%

(57 GPS stations). Furthermore, we calculate the PN spectral index and find that most of the PN spectral index approximates the FN, which indicates that the PN essence is similar to that of FN in Antarctica.

## 3 Results

### 3.1 GPS velocity field

After applying AIC noise analysis and ICA filters, we obtain a high-precision GPS velocity field, and then, we compare the

velocity changes with the raw GPS velocity. The result shows that the maximum difference is up to 1.15 mm yr$^{-1}$ (WWAY), the mean difference is 0.18 mm yr$^{-1}$, and 21 % (17 stations) of the velocities are greater than ±0.4 mm yr$^{-1}$. We exclude 9 stations that are inappropriate percentage statistics: FIE0, BUMS, MAW1, PECE, OHI2, STEW, VESL, MCM4, and HOOZ (processed GPS velocities are far greater than the raw velocities or the velocity directions are changed before and after applying AIC and ICA). We calculate the percentage of velocities that vary relative to raw GPS velocities, the maximum

variety of processed velocities is 80.22 %(ABBZ, which has a very small velocity magnitude), and the mean variety is 11.39 %. We find that the maximum velocity variety is up to 0.86 mm yr$^{-1}$, and the mean variety is 0.61 mm yr$^{-1}$ at the remaining 9 stations. Considering the elastic and GIA magnitudes, we cannot ignore these effects.

Figure 5 is the GPS velocity field after applying noise analysis and the AIC filter to Antarctica. The overall trend is upward. INMN has a maximum uplift velocity of 32.58 mm yr$^{-1}$, a mean velocity of 3.34 mm yr$^{-1}$ (TOMO were removed





because of some abnormal variations) (Martin-Espnol et al., 2016). Due to the lower upper mantle viscosity and mass loss caused by the collapse of the Larsen-B Ice Shelf (Nield et al., 2014), the north Antarctic Peninsula (NAP) mean uplift velocities (5.76 mm yr$^{-1}$) are larger than those of the SAP (3.72 mm yr$^{-1}$). The FRIS mean uplift velocities (4.73 mm yr$^{-1}$) are larger than those of the Ross Ice Shelf (ROSS, 0.74 mm yr$^{-1}$). The Amundsen Sea Embayment (ASE) has a mean uplift

velocity of up to 13.02 mm yr$^{-1}$, which is the maximum amount of ice mass loss (Groh et al., 2012). The most stable region is the East Antarctic (EA) coast, where the mean uplift velocity is only 0.07 mm yr$^{-1}$.

CAPF is located in NAP, and the vertical velocity is estimated at 15.0 $\pm$ 8.4 mm yr$^{-1}$ in Argus et al. (2014) based on approximately two years of GPS data, which is far greater than our estimated value of 4.07 $\pm$ 0.32 mm yr$^{-1}$ in this study. ROB4 is located on the west coast of the Ross Ice Shelf, and the vertical velocity is estimated at 1.14 $\pm$ 0.24 mm yr$^{-1}$, which

is similar to the 2.22 $\pm$ 3.20 mm yr$^{-1}$ estimated in Argus et al. (2014) based on approximately six years of GPS data and is dramatically different from the 7.54 $\pm$ 2.59 mm yr$^{-1}$ estimated in Thomas et al. (2011) based on 558 days of GPS data. These differences show that the GPS data time span plays an important role in the velocity estimation, and the longer the time span is, the more reliable the velocity estimation will be.

**3.2. Elastic correction**

In Antarctica, the GPS uplift velocities are dominated by the elastic deformation due to present ice mass loading and GIA. Riva et al. (2017) shown that the elastic response has a long wavelength influence in Antarctica; they used mass loss from glaciers, ice sheets, Greenland and Antarctic ice sheets in 1902 and 2014 to determine solid Earth deformation at regional and far fields. Based on the result in Riva et al. (2017), we calculated uplift velocities at 79 GPS sites. Figure 6 shows the GPS elastic velocities in Antarctica; the Antarctic Peninsula and ASE regions have larger elastic velocities and mean

magnitudes of 2.20 mm yr$^{-1}$ and 0.96 mm yr$^{-1}$, respectively. The FRIS and ROSS regions have smaller elastic velocities, while the EA has a negative elastic response. Clearly, the estimated GIA uplift rates would be significantly contaminated and in some areas dominated by neglecting the elastic response. When applying the elastic deformation correction, we consider GPS vertical velocities are mainly caused by GIA. We use the corrected GPS velocities to assess 7 GIA models: ICE-6G (VM5a), ICE-5G (VM2), WANG (CE-4G+RF3L20, β=0.4), W12a, Geruo13, IJ-05R2, and Paulson.

**3.3. GIA assessment**

To explore the application of GIA models in different regions, we divide Antarctica into 6 subregions(Sasgen et al. , 2013;Martin et al., 2016) and show these subregions in Fig.7. The station information is indexed in Table S1.

Notably, the reference frame origin of the GIA model is the center of mass (CM) of the solid Earth (CE), while the GPS velocities are estimated in the ITRF2008 reference frame, whose origin is the CM of the total Earth system. Argus et al.

(2014) thought the velocities between CM and CE caused by GIA are very small, but the velocities caused by the modern ice mass loss are more significant. If the ice loss in Greenland was 200 Gt yr$^{-1}$ and there is no ice loss in other areas, then the



velocity is approximately 0.22 mm yr$^{-1}$. The above corrections are much smaller than the uncertainty of the GIA models and GPS vertical velocities, and therefore, the impact of these corrections can be ignored in this study.

Figure 8 is the predicted uplift velocities of 7 GIA models, and the maximum, minimum, mean, and RMS values of the uplift velocities are listed in Table 2. From Fig.8, we can see that SAP, ASE, ROSS, and FRIS have larger uplift velocities,

which may be caused by the most ice mass loss since the LGM and the rapid response of the solid Earth (Martń-Espanol et al., 2016). The vertical velocities predicted over the West Antarctica (WA) are bigger than those in EA basins, while the vertical velocities have a smaller values along coastal EAs, and there are different sizes of subsidence areas in the interior across solutions, which may be because of the low upper mantle viscosities and higher values in EAs (An et al., 2015; vander Wal et al.,2015). Because of lacking ice history data in GIA models, EAs has a small variability between solutions (Martń-

Espanol et al., 2016). The spatial variability in all GIA models is larger than the GIA signal itself in many cases, especially in the interior areas of EAs where the mean GIA velocities are small. We find that the western margin of the Ross Ice Shelf, the ASE sector, the FRIS, and the Antarctic Peninsula(AP) have maximum variability.

The predictions of the ICE-5G, Geruo13 and Paulson models are quite similar in terms of spatial distribution, which may caused by the same ice model ICE-5G employed in the GIA modeling. The predictions are quite different among the ICE-6G,

WANG, IJ05-R2 and W12a models, which employed different ice models, indicating that the ice models play a major role in the predictions of GIA models. Earth models have much less effect than ice models in GIA modeling, which may be related to the unconsidered lateral variation in mantle viscosity (Ivins et al., 2005). ICE-6G, W12a and IJ05-R2 employed the new ice models; they have a similar distribution patterns of maximum uplift velocities and an obvious submerged trend in the interiors, while the magnitudes of the IJ05-R2 uplift velocities are much less than those of ICE-6G and W12a. From Table 2,

we can see that the IJ05-R2 velocities have the minimum standard deviation (std). The distribution pattern of the WANG model differs greatly from that of the other 6 GIA models; the pattern shows larger uplift velocities in the NAP and Enderby Land. All GIA models have maximum uplift velocities in the nearby ROSS and FRIS regions. ICE-6G has a peak descent velocity in the South Weddell Sea of approximately -2.20 mm yr$^{-1}$. The W12a has a peak descent velocity of approximately -6.11 mm yr$^{-1}$ near Coats Land; IJR5-R2 has no obvious peak descent velocity, which means that there is greater uncertainty

in some or all 3 GIA models, and systematic differences are also likely.

Generally, the Antarctic GIA models still have greater uncertainty with a lack of adequately accurate constraint data. As presented by the authors of Martń-Espanol et al. 2016, we use the weighted mean (WM), weighted root mean square (WRMS) and median values to evaluate the consistency between GPS vertical velocities and GIA model velocities. WM and WRMS are defined by formulas (6) and (7):

$$WM=\frac{\sum_{i=1}^{79}(p_i-O_i)w_i}{\sum_{i=1}^{79}w_i} \tag{6}$$

$$WRMS=\sqrt{\frac{\sum_{i=1}^{79}(P_i-O_i)^2w_i}{\sum_{i=1}^{79}w_i}} \tag{7}$$





where $P_i$ and $O_i$ are the GIA-modeled and GPS-observed velocities, and $W_i$ is the weight factor obtained by GPS measurement errors at each station:

$$w_i = \frac{1}{c_i\left(\sigma_i^{GPS}\right)^2} \quad i = 1, \cdots 79 \tag{8}$$

where $\sigma_i$ represents the measurement error at each GPS station and $c_i$ is calculated as follows:

$$C_i = \sum_{j=1}^{79} \exp\left(\frac{-d_{ij}}{I}\right) \tag{9}$$

x is distance matrix and $d_{ij}$ is the ith, jth value of the x relative to the 79 GPS locations, in order to deweight the sites that are near to other sites, we used the scale parameter I. Similar to Martín-Espanol et al. 2016, we also assume I = 250 km. The WM and WRMS results before and after applying the ICA and noise analysis are listed in Tables 3 and 4, respectively (* indicates the results of applying the ICA and noise analysis).

Table 3 shows the WMs of Antarctica and the subregions. After applying the ICA filter and noise analysis, the WM values of all GIA models are reduced in ASE. The ICE-6G, ICE-5G, WANG, W12a, and Geruo13 models are also reduced near FRIS. The WM values of most GIAs in other regions are increased. For all 79 stations, the WM of the residuals between the ICE-6G, WANG, W12a, and Paulson07 models and observed uplift velocities are increased. We think that the consistency between the raw GPS velocities and the 4 GIA model uplift velocities are overly optimistic, the two have poor consistency. The WM values of ICE-5G and Geruo13 are from negative to positive, which also indicates that the effects of the regional filter and the noise model are not negligible.

Table 4 shows the WRMSs of Antarctica and the subregions. The WRMS of the Antarctica peninsula (AP) and ASE are increased after applying the ICA filter and noise analysis, which we infer due to the local effects or inaccurate elastic model. In some regions with obvious GIA effects, such as the ROSS and FRIS regions (Argus et al., 2014; Martin et al., 2016), the WRMSs are effectively reduced. The WRMSs of all of Antarctica are reduced, which means that raw GPS velocities are affected by local effects. After applying the ICA filter and noise analysis, the local effects are depressed. In some regions with relatively good consistency between GPS observed velocities and GIA model predicted velocities, the consistency becomes better.

Figure 9 shows the summary statistics of WM and WRMS and the median values of residuals (GPS velocities with ICA filter applied and the ONM and GIA model predicted velocities). The WM of GPS and IJ05-R2 is -0.65 mm yr$^{-1}$, which indicates that the predicted velocities of IJ05-R2 are systematically smaller than GPS observed velocities. The WM of the other 6 GIA models range from 0.32 mm yr$^{-1}$ to 2.14 mm yr$^{-1}$, which means that the model predicted velocities are systematically larger than the GPS observed velocities. The WM of ICE-5G and Geruo13 are relatively small, which indicates that the two models are unbiased with GPS velocities. WANG has the maximum median and WM values. ICE-6G has the minimum WRMS, which we infer to be because the ICE-6G employed GPS data as a constraint (Argus et al., 2014).



## 4. Discussion

To evaluate the GIA models applicability in Antarctica, the estimates velocities and observed vertical velocities by an independent set of 79 GPS stations were compared. Figure10 is the discrepancies between the GIA velocities and GPS uplift rates at each GPS site; then, we perform the regional analysis and interpret the GIA uplift rates.

ASE: Figure 10 shows the difference between the GPS and GIA velocities at each GPS site, and the matching results are the worst in the ASE. The GPS uplift velocities along the ASE coast have larger differences, ranging from -1.82 mm yr[-1] (THUR) to 27.34 mm yr[-1] (BERP). All GIA model predicted velocities are systematically underestimated at the INMN, BERP, and BACK stations. ICE-6G has a maximum uplift velocity of approximately 7 mm yr[-1]. The next is W12a, which has an uplift velocity of approximately 5 mm yr[-1], and the other models are within 2 mm yr[-1]. From Table 4, we know that

the ASE region has the maximum WRMS, and the largest discrepancy between the GPS and GIA models is greater than 20 mm yr[-1] (INMN). Removing the INMN and BERP stations, which have large uplift velocities, reduces the WRMS values to 7.01 mm yr[-1], 5.86 mm yr[-1], 5.56 mm yr[-1], 5.92 mm yr[-1], 5.83 mm yr[-1], 5.52 mm yr[-1], and 5.62 mm yr[-1]. Seismic evidence reveals there is a very low upper mantle viscosity, about 1018 Pa s in this area (An et al., 2015; Heeszel et al., 2016), that could cause a fast response to ice mass changes at a smaller scale (Martin et al., 2016). The stations BACK, BERP and

TOMO are all located in the Pine Island Bay region, and recent studies indicate that fast ice mass loss occurs in both the Pine Island Glacier and Thwaites Glacier in this region. A sprawling network of low-lying canals, which is similar to a swamp, is hidden beneath the fast-flowing Thwaites Glacier. The abnormally large uplift velocities of the three stations reflect the elastic uplift caused by the modern ice mass loss.

ROSS: The GPS velocities are from -2.34 mm yr[-1] to 7.01 mm yr[-1] in the ROSS region, and the mean velocity is

approximately 0.72 mm yr[-1]. All GIA model predicted velocities are consistent with the GPS observed velocities. Except for the IJ05-R2 underestimation by 0.77 mm yr[-1], the other models overestimated the velocities in this region by approximately 1.03 mm yr[-1] ~ 2.24 mm yr[-1]. The W12a model has the maximum WRMS and overestimates by approximately 2.32 mm yr[-1], which suggests that the ice in W12a model was too much in LGM, or the upper mantle viscosity was too large (Martin et al., 2016).

AP: The GPS vertical velocities in the Antarctic Peninsula are generally larger than the predictions of all GIA models. This study's uplift estimate of the FONP station is 11.85 mm yr[-1], while the mean GIA prediction is 1.95 mm yr[-1]. One possible cause for such a difference is the crustal elastic response to the modern ice mass change. The Prince Gustav Ice Shelf and Larsen-A Ice Shelf collapsed in 1995. The neighboring Larsen-B Ice Shelf partially collapsed in 2002 and is quickly weakening and likely to completely disintegrate before the end of the decade. Ice shelves are the gatekeepers of

glaciers flowing from Antarctica toward the ocean (Martin et al., 2016). Without ice shelves, the glacial ice enters the ocean faster and accelerates the pace of global sea level rise. Thomas et al. (2011) found that the uplift velocities of the stations in this region increased obviously after the collapse of the Larsen-B Ice Shelf; for example, the velocity of PALM was 0.1 mm yr[-1] before 2002 and reached 8.8 mm yr[-1] after 2002. Except for WANG overestimating the velocities by 2.27 mm yr[-1], the





GIA models generally underestimated the velocities by more than 1.07 mm yr⁻¹~4.43 mm yr⁻¹ in the NAP. ICE-6G values are relatively consistent with the GPS velocities. The GPS vertical velocities in the NAP are generally larger than those in the SAP, which agrees with Wolstencroft et al. (2015), indicating a moderately low upper mantle viscosity in SAP, even though not as low as NAP.

FRIS: FRIS is near the Weddell Sea Embayment, the crustal thickness in the transition between EAs and WAs and the mantle viscosity are moderate (An et al., 2015; Heeszel et al., 2016). The mean GPS uplift velocity is 4.28 mm yr⁻¹, the uplift velocities are underestimated by the ICE-5G, Geruo13, and IJ05-R2 models by 3.49 mm yr⁻¹, 3.45 mm yr⁻¹ and 0.88 mm yr⁻¹, respectively, and overestimated by 0.38 mm yr⁻¹ ~1.65 mm yr⁻¹ by the other models. The matching results between the GPS and GIA are better overall, so we think that the uplift is mainly caused by the GIA in this region, which agrees with the

findings of (Arguset al.,2014; Martin et al.,2016).

    EAs: Our GPS vertical velocities along the EAs coast range from -1.94 mm yr⁻¹ ~2.5 mm yr⁻¹ and are smaller than those in WAs The GIA model velocities agree with the GPS velocities. The uplift velocities are underestimated by ICE-5G and Geruo13 by approximately 0.49 mm yr⁻¹ and 0.53 mm yr⁻¹, respectively, and overestimated by the other models by 0.38 mm yr⁻¹ ~1.65 mm yr⁻¹. The basement of EAs is an ancient craton, and the geological structure is very stable, so there is no

significant geological activity occurring in this region. The vertical movements along the EAs coast are all nonsignificant, showing the effects of the GIA or recent ice and snow accumulations to be small. GRACE gravity data from 2009-2013 show that the coast of Queen Maud Land in EAs accumulated ice and snow at a rate of 150 Gt yr⁻¹ (Argus et al., 2014). The precipitation data from 2009-2012 also measure fast accumulation, but the accumulation from 1980-2008 is approximately zero, indicating that the recent ice and snow accumulation is anomalous and represents interannual variations (Boening et al.,

2012). Overall, there is no significant geological activity in EAs, and the effects of the GIA and ice mass loading are small in this region.

## 5. Conclusions

High-precision GPS data are an effective approach for studying regional crustal displacements. Studying the regional crustal displacement in Antarctica has important value as a reference for the formation and evolution of global plate tectonics in

addition to creating and maintaining reference frames and monitoring the dynamics of ice and snow in polar regions. For the regions of Antarctica with complex terrain, we removed the CME of the residual time series by ICA filtering of the time series recorded at 79 GNSS stations in Antarctica, and then, the AIC is used to determine the ONM. Finally, we used high-precision GPS data to assess the 7 GIA models. The results are as follows:

1.    After applying an AIC noise analysis and the ICA filter, the maximum difference is up to 1.15 mm yr⁻¹, the mean

difference is 0.18 mm yr⁻¹, 21 % (17 stations) of the velocities are greater than ±0.4 mm yr⁻¹, the maximum variety of processed velocities is 80.22 %, and the mean variety is 11.39 %.



2.  After applying the ICA filter and noise analysis, the WM values of all GIA models are reduced in ASE, and the ICE-6G, ICE-5G, WANG, W12a, and Geruo13 models are also reduced near FRIS; the WM values are increased for most of the GIA in other regions; for all 79 stations, the WMs of residuals between the ICE-6G, WANG, W12a, and Paulson07 models and observed uplift velocities are increased. The WRMS of AP and ASE are increased after applying the ICA filter and noise analysis; in some regions with obvious GIA effects, such as the ROSS and FRIS regions, the WRMSs are effectively reduced. The WRMSs of all of Antarctica are reduced, which means that the raw GPS velocities are affected by local effects. After applying the ICA filter and noise analysis, the local effects are depressed; in some regions with relatively good consistency between the GPS observed velocities and GIA model predicted velocities, the consistency becomes better.

3.  The predicted velocities of IJ05-R2 are systematically smaller than the GPS observed velocities; the other 6 GIA model predicted velocities are systematically larger than the GPS observed velocities. The WMs of ICE-5G and Geruo13 are relatively small. WANG has the maximum median and WM values. ICE-6G has the minimum WRMS. Because the upper mantle viscosities in the NAP are lower than in the SAP, the GPS velocities shows the largest vertical deformation in the NAP than SAP. In the FRIS ice shelves, the observed GPS velocities and the predicted GIA model velocities are consistent. In EA, the vertical motion is nonsignificant, and the GIA and ice loading have a small impact in this area.

**Author contributions**

Wenhao Li and Shengkai Zhang conceived and designed the experiments; Wenhao Li, Jintao Lei and Qingchuan Zhang analyzed the data, Fei Li gave the critical suggestions, Wenhao Li wrote the main manuscript text, Feng Xiao and Lexian Yuan  helped with the writing of text.  All authors declare that they have no conflicts of interest.

**Competing interests**

The authors declare that they have no conflicts of interest.

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

**Caption of Figures**

**Figure 1. The distribution of Global Positioning System (GPS) stations in Antarctica**

**Figure 2. The results of IC1-IC8 components (the black arrows are a positive spatial response, the red are a negative spatial response)**

**Figure 3. the residual time series (left) and raw time series (right) of GMEZ before and after regional filter using the ICA (blue lines are the raw time series and the orange are filtered time series).**

**Figure 4. The RMS values of the residual time series before and after applying the ICA filters (%)**

**Figure 5. The GPS velocity field after applying noise analysis and AIC filter**

**Figure 6. Elastic velocity of GPS in Antarctic (mm yr$^{-1}$)**




**Figure 7. The divided 6 subregionals in Antarctic. Red represent those within the North Antarctic Peninsula; yellow those within the South Antarctic Peninsula; balck those within the Amundsen Sea Embaymen; blue those within the Filchner-Ronne Ice Shelf; light green those within the Ross Ice Shelf; and brown those along coastal East Antarctica.**

**Figure 8. The uplift velocities of GIA models (mm yr⁻¹)**

5 **Figure 9. The summary statistics WM and WRMS, Median values are indicated in brackets (mm yr⁻¹)**

**Figure 10. The discrepancies between the modeled and the observed GIA uplift rates estimated from different solutions computed at each GPS site (Red circles indicate places where the estimated GIA rates underestimate the observed velocities from GPS; blue circles indicate the converse.)**

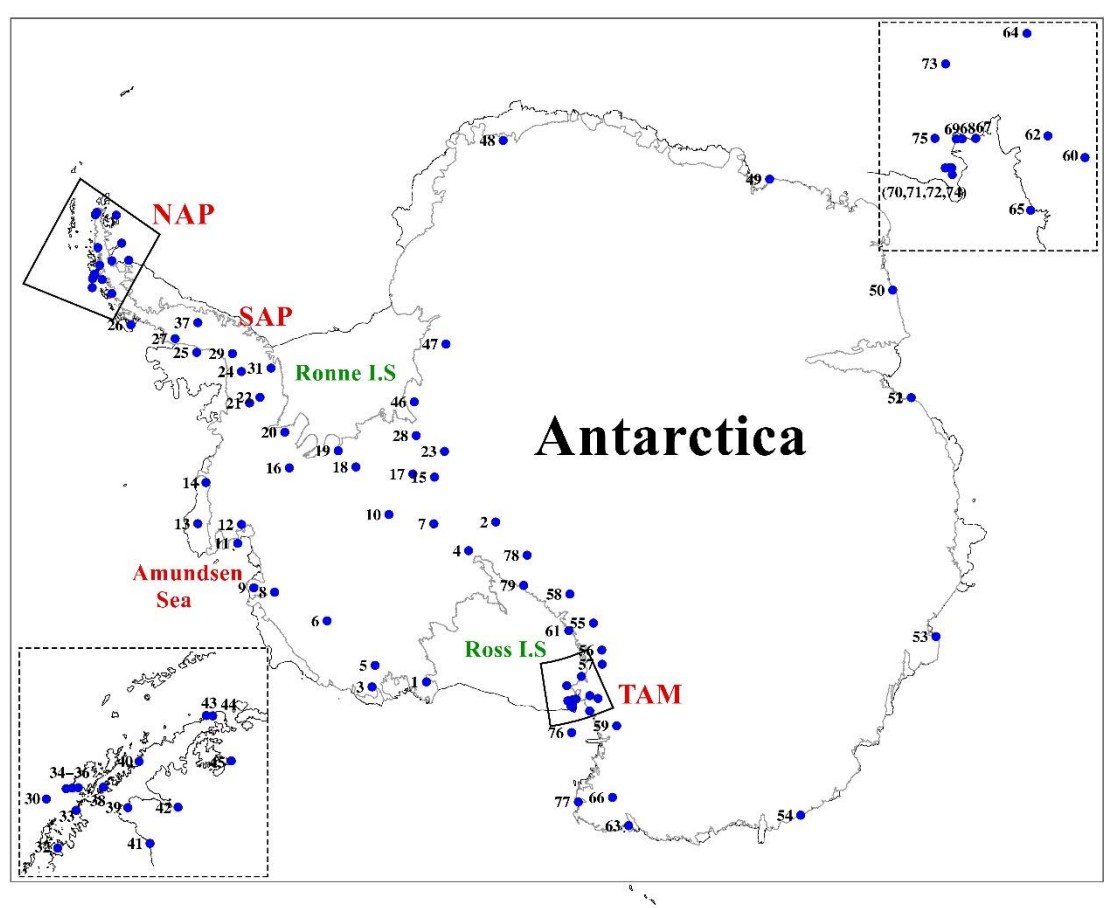

**Figure 1**. The distribution of Global Positioning System (GPS) stations in Antarctica









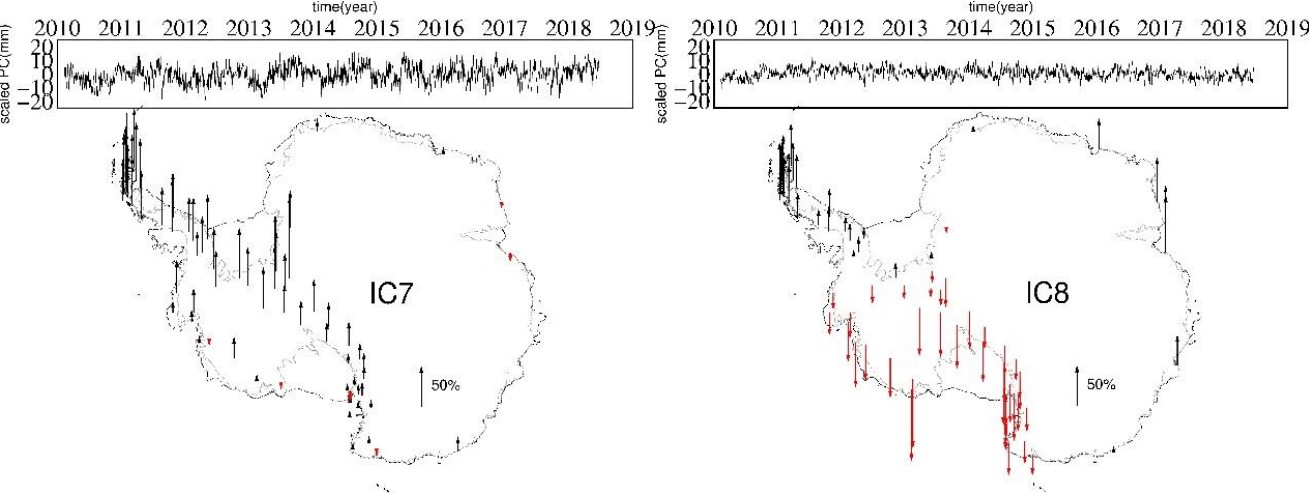

**Figure 2.** The results of IC1-IC8 components (the black arrows are a positive spatial response, the red are a negative spatial response)

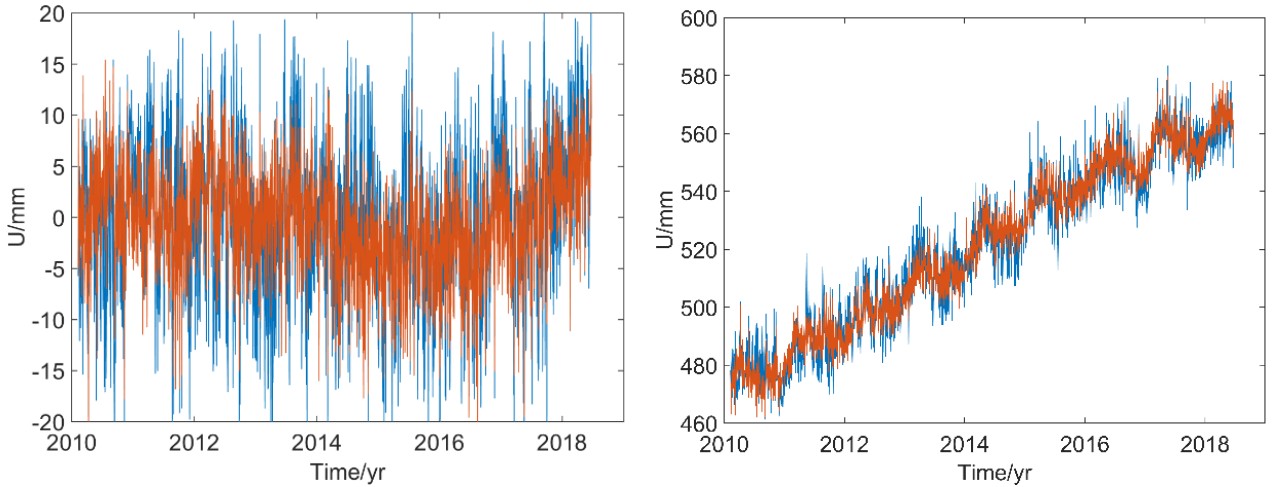

5    **Figure 3.** the residual time series (left) and raw time series (right) of GMEZ before and after regional filter using the ICA (blue lines are the raw time series and the orange are filtered time series).



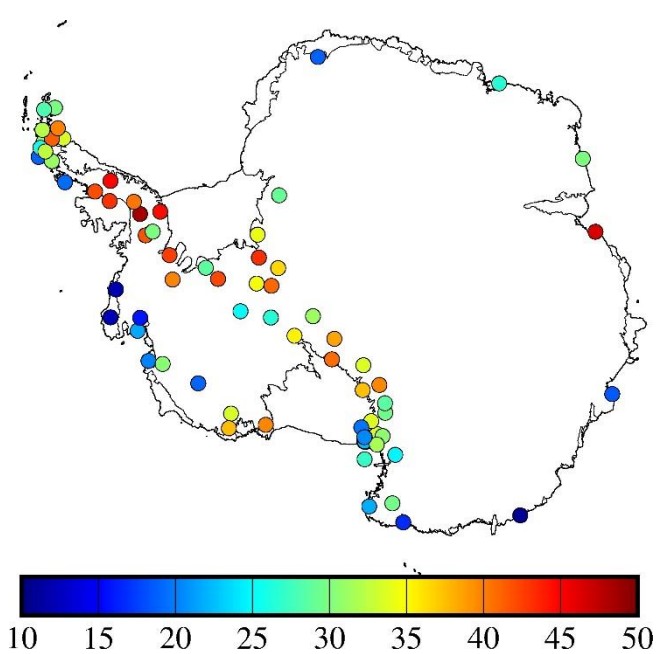

**Figure 4.** The RMS values of the residual time series before and after applying the ICA filters (%)

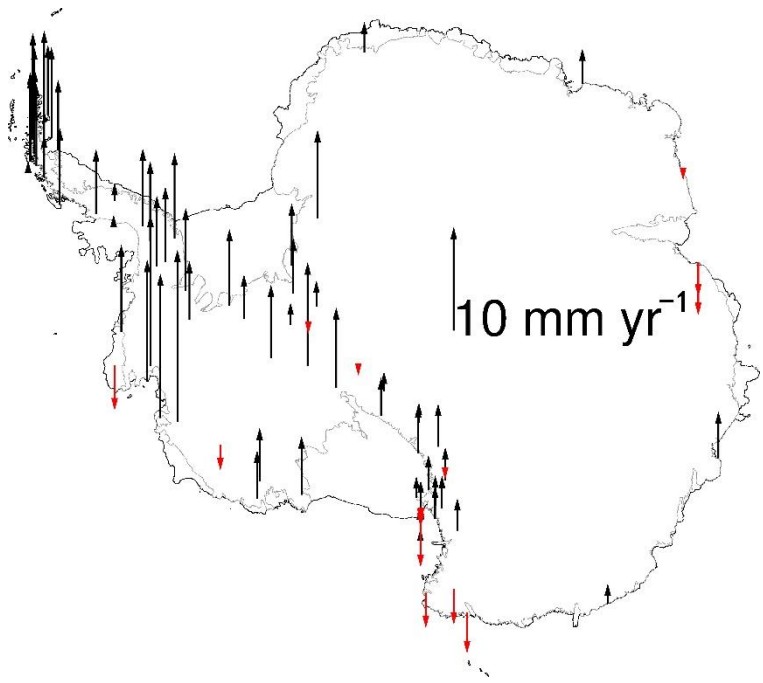

5                          **Figure 5**. The GPS velocity field after applying noise analysis and AIC filter





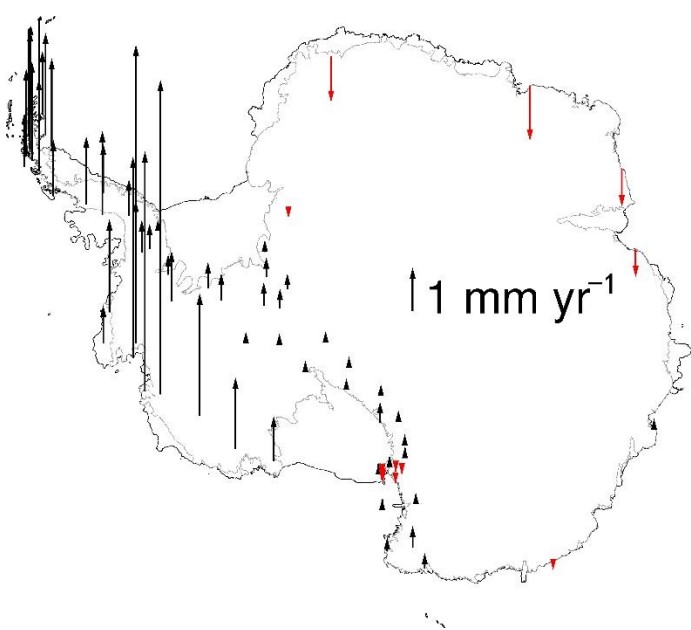

**Figure 6**. Elastic velocity of GPS in Antarctic (mm yr$^{-1}$)

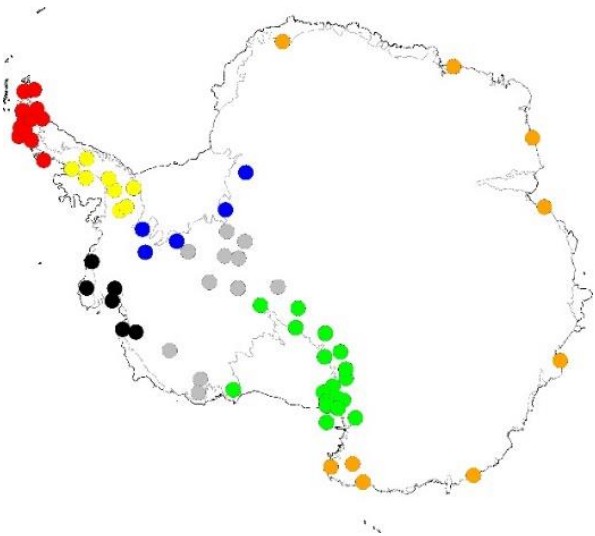

5  **Figure 7.** The divided 6 subregionals in Antarctic. Red represent those within the North Antarctic Peninsula; yellow those within the South Antarctic Peninsula; balck those within the Amundsen Sea Embaymen; blue those within the Filchner-Ronne Ice Shelf; light green those within the Ross Ice Shelf; and brown those along coastal East Antarctica.





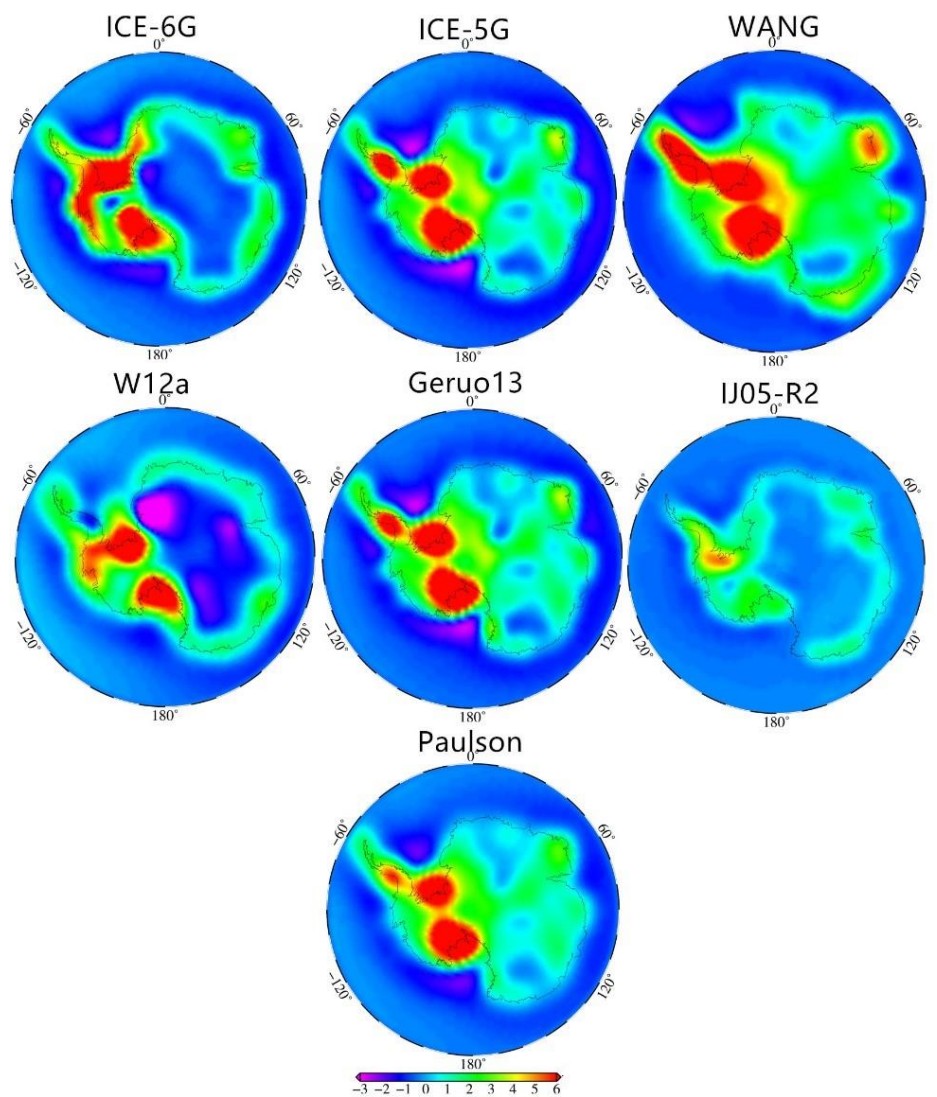

**Figure 8.** The uplift velocities of GIA models(mm yr$^{-1}$)

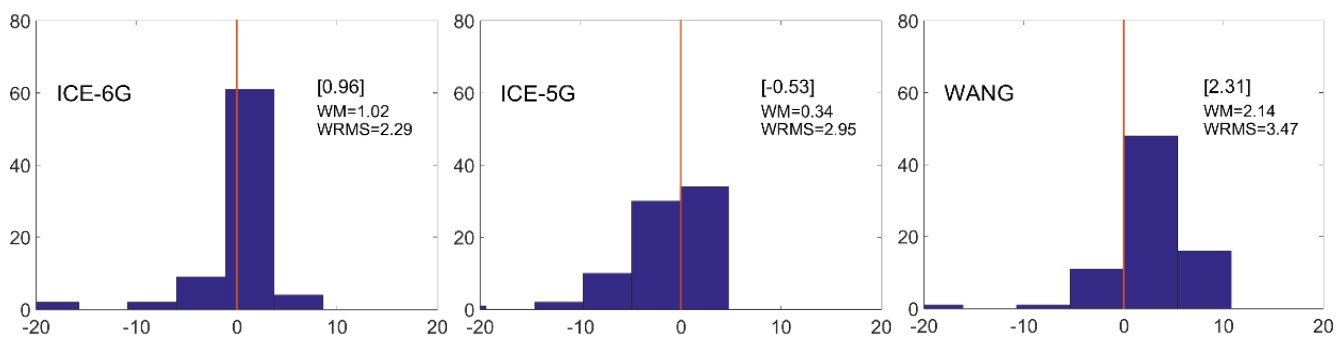




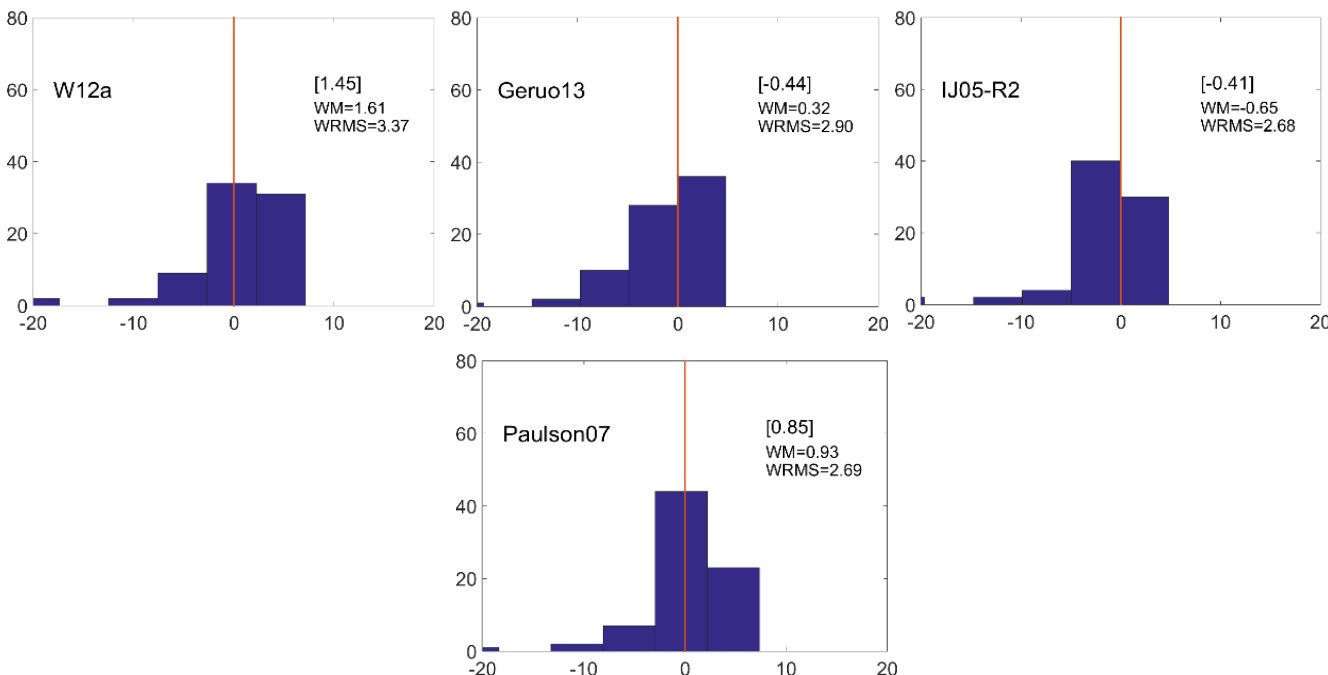

**Figure 9.** The summary statistics WM and WRMS, Median values are indicated in brackets (mm yr$^{-1}$)



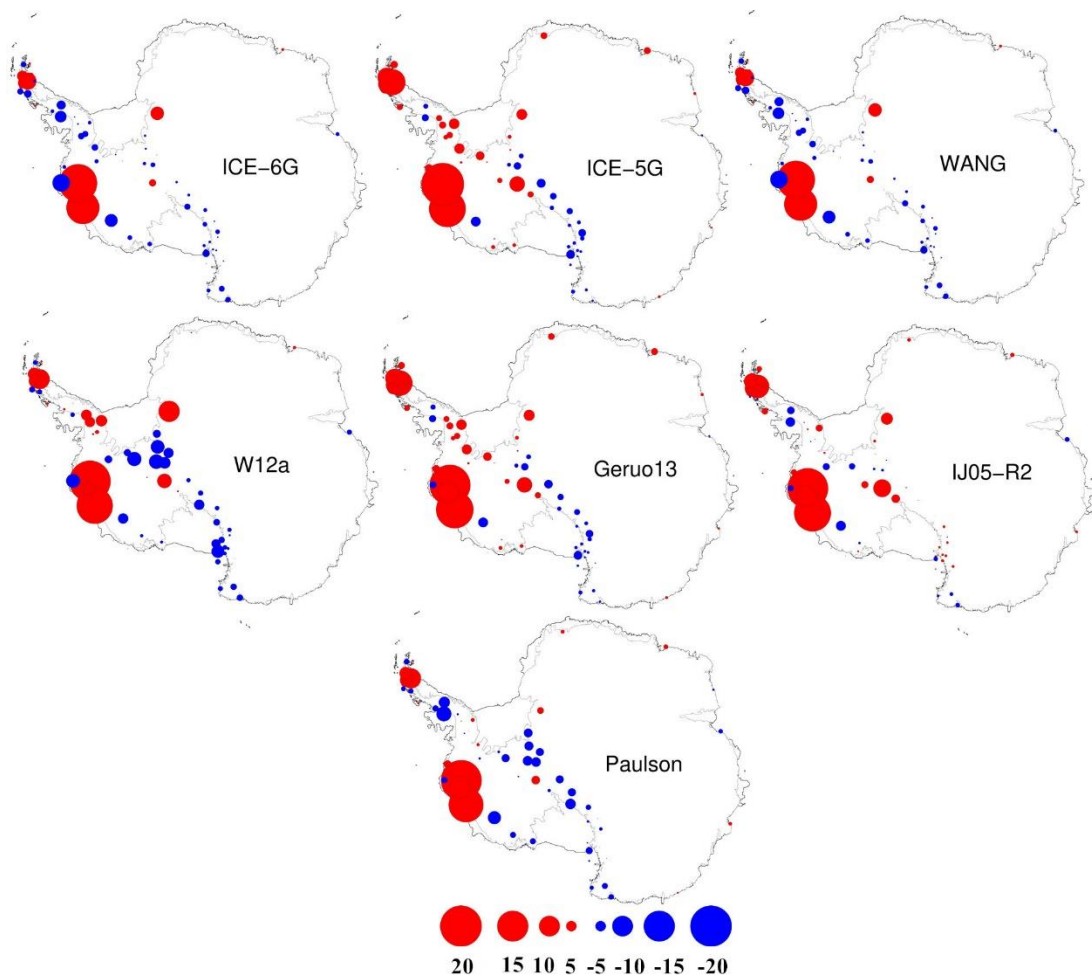

**Figure 10.** The discrepancies between the modeled and the observed GIA uplift rates estimated from different solutions computed at each GPS site (Red circles indicate places where the estimated GIA rates underestimate the observed velocities from GPS; blue circles indicate the converse.)

## 5  Tables

Table 1. The ONMs and corresponding velocities of GPS stations (mm yr$^{-1}$)

| Sites | Noise Model | Velocity | Sites | Noise Model | Velocity |
|-------|-------------|----------|-------|-------------|----------|
| ABBZ | WN+PL | -0.11±0.391 | MCM4 | WN+PL | -0.42±0.157 |
| BACK | WN+FN | 15.82±0.923 | MCMD | WN+PL | 0.12±0.218 |
| BENN | WN+PL | 9.91±0.167 | MIN0 | WN+PL | 0.64±0.124 |



| | | | | | |
|---|---|---|---|---|---|
| BERP | WN+PL | 27.34±0.657 | MKIB | WN+FN | 6.19±0.514 |
| BRIP | WN+PL | 1.11±0.236 | OHI2 | WN+PL | 1.13±0.514 |
| BUMS | WN+FN | 0.57±0.477 | OHI3 | WN+PL | 1.89±0.637 |
| BURI | WN+PL | 1.15±0.109 | PAL2 | WN+PL | 6.18±0.330 |
| CAPF | WN+PL | 4.07±0.318 | PALM | WN+PL | 6.20±0.300 |
| CAS1 | WN+PL | 1.90±0.390 | PALV | WN+PL | 6.90±0.344 |
| CLRK | WN+PL | 2.45±0.107 | PATN | WN+PL | 2.87±0.267 |
| COTE | WN+PL | 1.02±0.131 | PECE | WN+FN | 0.81'±0.548 |
| CRAR | WN+PL | 0.95±0.177 | PHIG | WN+RW+GGM | -2.35±1.618 |
| CRDI | WN+PL | 3.26±0.142 | PIRT | WN+PL | 1.81±0.181 |
| DAV1 | WN+PL | -1.20±0.148 | PRPT | WN+PL | 1.62±0.797 |
| DAVE | WN+PL | -2.36±0.166 | RAMG | WN+FN | 1.43±0.237 |
| DEVI | WN+PL | 2.01±0.098 | RMBO | WN+PL | 2.58±0.239 |
| DUM1 | WN+FN | 0.59±0.785 | ROB4 | WN+PL | 1.14±0.249 |
| DUPT | WN+RW+GGM | 10.88±1.269 | ROBN | WN+PL | 7.20±0.394 |
| FALL | WN+PL | 5.40±0.195 | ROTH | WN+PL | 4.67±0.637 |
| FIE0 | WN+PL | 0.19±0.470 | SCTB | WN+PL | -0.29±0.148 |
| FLM5 | WN+PL | 1.12±0.097 | SDLY | WN+PL | -0.79±0.156 |
| FONP | WN+PL | 15.03±0.357 | SPGT | WN+PL | 10.89±0.594 |
| FOS1 | WN+PL | 0.27±0.423 | STEW | WN+FN | 0.67±0.601 |
| FTP4 | WN+PL | 1.25±0.135 | SUGG | WN+FN | 4.73±0.468 |
| GMEZ | WN+PL | 5.35±0.342 | SYOG | WN+PL | 1.26'±0.375 |
| HAAG | WN+PL | 5.80±0.272 | THU4 | WN+PL | -1.82±0.574 |
| HOOZ | WN+PL | -0.31±0.45 | TOMO | WN+FN | 52.63±1.032 |
| HOWE | WN+FN | -0.30±0.379 | TRVE | WN+PL | 3.49±0.327 |
| HOWN | WN+PL | 2.89±0.188 | VESL | WN+PL | 1.01±0.746 |
| HUGO | WN+FN | 0.31±0.785 | VL01 | WN+PL | -1.24±1.111 |
| IGGY | WN+FN | 0.59±0.328 | VL12 | WN+PL | -1.25±0.655 |
| INMN | WN+FN | 32.58±1.114 | VL30 | WN+FN | -1.58±1.833 |
| JNSN | WN+PL | 5.07±0.433 | VNAD | WN+PL | 5.63'±0.395 |



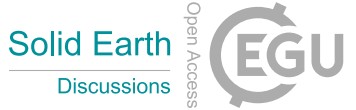

| LNTK | WN+FN | 4.18±0.817 | WHN0 | WN+PL | 0.28±0.194 |
| LPLY | WN+FN | 6.56±1.083 | WHTM | WN+RW+GGM | 4.46±1.472 |
| LWN0 | WN+PL | 1.70±0.072 | WILN | WN+PL | 4.91±0.446 |
| MACG | WN+FN | 0.19±0.542 | WLCH | WN+PL | 0.52'±0.421 |
| MAW1 | WN+PL | -0.13±0.138 | WLCT | WN+FN | -0.15±0.573 |
| MBIO | WN+PL | 3.83±0.543 | WWAY | WN+RW+GGM | 6.80±2.776 |
| MCAR | WN+PL | 2.07±0.186 | | | |

**Table 2.** The maximum, minimum, mean values and standard deviation of 7 GIA models uplift velocities(>60 ˚S)

| GIA | Max( mm yr$^{-1}$) | Min( mm yr$^{-1}$) | Mean( mm yr$^{-1}$) | Std( mm yr$^{-1}$) |
|---|---|---|---|---|
| ICE-6G | 13.50 | -2.20 | 0.71 | 1.15 |
| ICE-5G | 13.90 | -2.80 | 1.58 | 1.10 |
| WANG | 15.27 | -2.13 | 2.60 | 1.15 |
| W12a | 10.33 | -6.11 | 0.58 | 0.97 |
| Geruo13 | 15.00 | -2.70 | 1.34 | 1.19 |
| IJ05-R2 | 5.24 | -0.88 | 0.22 | 0.45 |
| Paulson | 12.46 | -1.98 | 1.50 | 1.07 |

5    **Table 3.** The WM results in North Antarctic Peninsula (NAP), South Antarctic Peninsula (SAP), Amundsen Sea Embayment (ASE), Margins of the Ross (ROSS), Filscher-Ronne Ice Shelves(FRIS), and East Antarctica (EA)a

| Model | NAP(15) | | SAP(8) | | ASE(5) | | ROSS(25) | | FRIS(5) | | EA(8) | | Antarctica(79) | |
|---|---|---|---|---|---|---|---|---|---|---|---|---|---|---|
| | WM | WM* | WM | WM* | WM | WM* | WM | WM* | WM | WM* | WM | WM* | WM | WM* |
| ICE6G | -1.18 | -1.49 | 2.57 | 2.42 | -4.68 | -3.78 | 1.25 | 1.16 | 1.92 | 1.38 | -0.09 | 0.49 | 0.48 | 1.02 |
| ICE5G | -4.80 | -5.21 | -1.71 | -1.14 | -9.98 | -9.04 | 1.35 | 1.58 | -2.72 | -0.34 | -1.03 | -0.57 | -1.04 | 0.34 |
| WANG | 2.28 | 1.98 | 4.22 | 5.29 | -8.72 | -7.81 | 1.89 | 1.46 | 2.62 | 2.46 | 1.76 | 2.47 | 1.78 | 2.14 |
| W12a | -1.91 | -2.27 | -1.98 | -2.22 | -6.27 | -5.42 | 2.37 | 2.06 | 2.75 | 2.11 | 0.39 | 0.93 | 0.80 | 1.61 |
| Geruo13 | -4.51 | -4.91 | -1.64 | -1.07 | -9.89 | -8.95 | 1.33 | 1.52 | -2.65 | -0.32 | -1.12 | -0.63 | -1.01 | 0.32 |
| IJ05R2 | -3.71 | -4.06 | 0.25 | 0.59 | -10.02 | -9.14 | -0.86 | -0.91 | 0.51 | 0.73 | -0.16 | 0.65 | -1.38 | -0.65 |
| Paulson | -2.04 | -2.40 | 1.65 | 2.33 | -10.06 | -9.16 | 1.07 | 0.72 | 1.45 | 1.70 | 0.05 | 1.01 | 0.10 | 0.93 |

[a] The number of sites in each area is indicated in parentheses; * represents the results of applying ICA filtering and AIC noise analysis.





**Table 4.** The WRMS results in North Antarctic Peninsula (NAP), South Antarctic Peninsula (SAP), Amundsen Sea Embayment (ASE), Margins of the Ross and Filscher-Ronne (FRIS) Ice Shelves, and East Antarctica (EA)[a]

| Model | NAP(15) | | SAP(8) | | ASE(5) | | ROSS(25) | | FRIS(5) | | EA(8) | | Antarctica(79) | |
|---|---|---|---|---|---|---|---|---|---|---|---|---|---|---|
| | WRMS | WRMS* | WRMS | WRMS* | WRMS | WRMS* | WRMS | WRMS* | WRMS | WRMS* | WRMS | WRMS* | WRMS | WRMS* |
| ICE6G | 3.02 | 3.31 | 2.89 | 2.96 | 11.20 | 11.83 | 1.47 | 1.35 | 2.37 | 2.22 | 0.87 | 1.10 | 3.49 | 2.29 |
| ICE5G | 5.72 | 6.14 | 2.86 | 2.97 | 14.02 | 14.22 | 1.98 | 1.92 | 3.16 | 2.52 | 1.37 | 1.29 | 4.61 | 2.95 |
| WANG | 3.62 | 3.57 | 5.32 | 6.35 | 13.28 | 13.55 | 2.55 | 1.93 | 4.09 | 3.99 | 2.94 | 3.14 | 4.70 | 3.47 |
| W12a | 3.53 | 3.88 | 2.85 | 3.46 | 11.90 | 12.37 | 2.91 | 2.67 | 3.59 | 3.01 | 0.99 | 1.49 | 4.27 | 3.37 |
| Geruo13 | 5.47 | 5.88 | 2.81 | 2.93 | 13.96 | 14.16 | 1.93 | 1.85 | 3.09 | 2.50 | 1.45 | 1.32 | 4.54 | 2.9 |
| IJ05R2 | 4.76 | 5.14 | 1.80 | 2.35 | 14.13 | 14.26 | 1.32 | 1.22 | 2.15 | 2.32 | 1.10 | 1.45 | 4.24 | 2.68 |
| Paulson | 3.58 | 3.96 | 2.88 | 3.61 | 14.14 | 14.30 | 1.64 | 1.16 | 2.58 | 2.83 | 1.40 | 1.54 | 4.22 | 2.69 |

[a] The number of sites in each area is indicated in parentheses; * represents the results of applying ICA filtering and AIC noise analysis.