# Peer review of "An assessment of GIA solutions based on high-precision GNSS velocity field for Antarctica"

_Solid Earth, 2019_

## Referee Comment (RC1) · Matt King (Referee) · 19 Aug 2019

The authors present an analysis of GPS time series with a focus on common-mode filtering and its effects on derived velocity fields. It then compares the results vertical velocities to a range of GIA models. This is a worthy topic of investigation.

I am concerned that the manuscript in its present form does not present a clearly robust advance on the work already published in the literate, some of which is cited in the manuscript.

1. The work is in part of update of the work of Liu et al and Martin-Espanol et al. Liu et al consider ICA in the context of Antarctic vertical velocities and compare the resulting velocities to GIA models. Martin-Espanol et al do the same but without the filtering.

The authors do not sufficiently engage with these papers to explain the advance of their work. The Liu et al work is mentioned in a single sentence in the introduction only despite it being so similar. I think the main advance is longer time series but there could be other things, but I also think there are some backward steps (see below). In general, the introduction does not contain a complete review of people who have worked on the testing of GIA models in Antarctica and so does not set its own unique contribution in context.

2. the robust comparison of GPS velocities and GIA models requires robust consideration of elastic deformation. The authors uses the grids of Riva et al. for this purpose. This approach is problematic due to 1) that product not overlapping fully in time with the authors' GPS time series; 2) the authors appearing to apply the correction as a time-constant rate whereas elastic deformation is potentially highly nonlinear; and 3) the Riva et al product being explicitly designed for *far* field studies since the input data sets are low resolution (from GRACE in the case of Antarctica), As such, the product cannot accurately represent changes over spatial scales <300km. This is a backward step compared to the approach of Martin-Espanol (although there is room for further improvement in this regard). As such, I do not think the GPS velocities, after correction for elastic effects, can be compare to GIA models robustly.

3. The discussion and context misses one key recent publication, that of Barletta et al in Science where they show that the Amundsen Sea embayment is likely underlain with low viscosity mantle and hence is sensitive to very recent load changes only. The discussion of the northern Antarctic Peninsula does not consider the work of Nield et al 2014 even though that work is cited elsewhere in the manuscript. There are instances where discussion or conclusions are made which are not new or from the present work. For instance, p9 lines 5-24 are either out of date or repeat points made in the literature previously (also p11 Line 10-16 and abstract line 22-24)).

4. some of the methods are not fully described. The authors do not describe when they estimate offsets in the timeseries (due to, for example, equipment changes) - the

methods (line 21 p 3) suggests just a single constant offset is estimated for each site - if this is so, this would be a major methodological error.

5. in common mode filtering there is a chance to remove some trends or accelerations by accident. The authors do not test if their ICAs show linear or long-period accelerations which could remove important velocity. IC3 seems to show something along these lines. The authors also do not try and relate the ICAs to any physical or systematic errors (this point not being critical but it does mean the filtering is very blind and just a mathematical application, which can be dangerous).

6. Section 2.1 says the authors remove offsets, annual and semi-annual terms before further analysis - this sound slike it is before their later HECTOR analysis - in which case the uncertainties from HECTOR will be under-estimated - they should be estimated within HECTOR.

The English is quite good but there are locations where it needs editing to make sure the meaning is intended. I indicate some of these.

Minor remarks: P1L1: GNSS in title but GPS throughout all the paper (and only GPS used in the analysis by NGL) P1L11: past changes in mass loading L13: deformation of the crust is the changing shape L20 and throughout: specifying velocities to 0.01mm/yr is not warranted by the uncertainties. suggest 0.1mm/yr L22: the WANG model does not over-predict but for the wrong reason (c.f Nield et al 2014) L24: Filchner L28: delete slower. Not clear how it influences "plate tectonics" L30: English

P2L9: delete "considering ...;" P13: effaciously - wrong word L15: "on a spatial scale" - English L19: not enough information to understand why Gaussian distribution is relevant L21: new paragraph at "Relative to PCA" L27: these are good refs for introducing the coloured nosie, but they are analysis of very old data - suggest something new like Santamaria-Gomez or work of Klos L31: what does complex terrain have to do with it? L31-32: English

P3L8: the two variants of IJ05R2 are described but only one is shown in the later analysis (which one?), L17: Could do with a summary (and reference to relevant paper of Blewitt or Kreemer etc) to the analysis. Note that the analysis uses the GMF for tropospheric zenith delay which makes the analysis sub-state of the art but probably not critical here (it will come into the common mode noise) L21: Argus et al 2014 and Wolstencroft et al mention the effect of ice or snow on the positions. These mainly affect the horizontal signals - I presume your editing here just considered the vertical component of hte coordinate? P4L8: the simulation approach is not clear and needs further description. I think the ICAs include coloured noise but did these simulations? What is the impact?

L17: the definition of 'residual' is not clear. I think this is different to the residual series in Section 2.1 L27: Bos is not an appropriate reference for AIC. Why not BIC also?

P5L15: is the comparison of before/after filtering using the same noise model or estimating a new noise model each time? L17: the approach to rejecting stations is not clear L20: here and after "variety" is not the right word. variation? change? Not clear what is meant by "Considering the elastic ... effects."

P6L5: Barletta et al Science 2018 is needed to be considered L7-14: needs an introduction to explain why these stations are discussed. Are these the ones with large differences to previous works? I presume these results are from teh filtered results? But is the difference due to the different data span or the filtering or both?

P7L1: see also the work of Schumacher et al GJI on the difference between ITRF2008 CM and GIA model CE L26: "greater uncertainty" - than what? L27: delete "the authors of"

P8L9: in what follows there seems to be a lot of discussion of the unfiltered results (discussing WM not WM* for instance)

P9L3: 79 stations was reduced to anotehr number due to rejecting some sites L14-

18: these statemetns need references. The mention of Thwaites having sub-glacial channels is not clearly related to the actual paper. L32: Nield et al 2014 is curiously missing here. Zhao et al EPSL is also missing for southern Peninsula. Nield et al 2016 GJI is missing for Ross Sea region

P10L16: the variation in accumulation could be important but the authors do not describe why. These should, in principle, be corrected in a robust elastic model

References:Wang reference has an issue with journal of geodynsmics and Nature Geoscience listed Nield et al reference needs space between rapid and bedrock Geruo et al suggests the author's family name is Geruo - it is actually "A" so this should be A et al.

Figure 2: it would be good to see the time series at a larger scale so we can see the detail.

Figure 4,8: the rainbow colour scale is now regarded as misleading - search online for #endrainbow

Figure 5: some of the smaller arrows are hard to see. Are coloured circles more useful?

Table 2: which 6G model? _C? Note there was a bug in some of the calculations for ICE6G but they were updated on Peltier's website

Matt King, Aug 19, 2019

---

## Referee Comment (RC2) · Anonymous Referee #2 · 21 Aug 2019

The influence of common mode error(CME) and noise model on GPS accuracy is analyzed in this study using more than 79 stations with long-time series (around 9 years). The applicability of some GIA models is also accessed using the derived GPS velocity field, which shows that CEM and noise models are not negligible. GIA influences the plate tectonics, crustal displacements, geoid, etc. Differences in predictions of GIA for Antarctica exist due to the uncertainties in de-glacial history and Earth rheology. Hence, it is necessary to determine the applicability of different GIA models. However, the actual GPS velocities are somewhat affected by the CME and the optimal noise model. In general, this manuscript addresses an interesting topic and has significant contribution in this research field. Hence, I recommend its publication after considering the following minor comments.

[Figure]

MinorÂăcomments: (1) The elastic model does not overlap fully in time with the GPS time series, will it influence the results? (2) The introduction needs more complete review of the assessment GIA models. (3) Can you more precisely describe the noise models before and after filtering? (4) GNSS and GPS are inconsistency, GNSS in title but GPS throughout all the paper. (5) P6L7-14: the discussed stations need some introductions to explain further. Are these stations with large differences to previous results? (6) P4L17: the definition of 'residual' is same as the residual series in Section 2.1? (7) Figure 2 and Figure 5 are not clear, difficult to see the details. It is better to expand the scales.

---

## Author Comment (AC1) · 1 Oct 2019

Dear Reviewers,

  We are truly grateful for the critical comments and thoughtful suggestions provided by you. Based on these comments and suggestions, we have made careful modifications to the original manuscript. All changes made to the text are marked in red font. The main corrections in the paper and the responses to your comments are listed below. We are at your disposal for any further information and willing to improve further our manuscript by adding the considerations provided in our reply. Kind regards.

(1) The elastic model does not overlap fully in time with the GPS time series, will it influence the results?

**Response:** The elastic correction needs a high-resolution model of surface mass variation, and then the mass changes are converted into displacements using Green's functions. However, because of the lack of observed data and the data fusion problem, accurately quantifying elastic corrections are difficult. In this paper, we assume that the elastic velocity remains constant for nearly 20 years. The elastic model of Riva et al. (2017) is used to compute the elastic velocity to explore the noise and CME effects on GIA assessments. Then, the effects on GPS velocity estimates are revealed, with the results providing a reference for future research.

(2) The introduction needs more complete review of the assessment GIA models.

**Response:** We have added the relevant content:

"Mart ń-Espa ñol et al. (2016) used the elastic-corrected GPS vertical velocities in Antarctica over the period 2009-2014 to asses 8 GIA models, including forward and inverse methods, and they found systematic underestimations of the GPS rates over specific regions characterized by low mantle viscosities and thin lithosphere. Liu et al. (2018) applied ICA and PCA for 53 GPS stations from 2010 to 2014 and used the white noise plus power law (PL) noise model to estimate GPS velocities, and after correction for elastic effects, they assessed the consistency among 4 GIA models and GPS vertical velocities. They that the consistency between the GPS observed velocities and GIA models were generally improved after spatiotemporal filtering. Mart ń-Espa ñol et al. (2016) and Liu et al. (2018) used 53 GPS stations' velocities to assess the GIA models, although Mart ń-Espa ñol et al. (2016) did not perform filtering, and both studies considered only one noise model. A uniform criterion is not available to judge the effects of CME and noise models; therefore, a quantify study is needed of the effects based on GPS velocity estimates and GIA assessments. In this paper, we used more than 79 stations with long time series (around 9 years) to achieve an accurate velocity, and then the influence of common mode error (CME) and 5 noise models on the GPS accuracy was analyzed. Finally, we assessed the application of GIA models in Antarctica."

(3) Can you more precisely describe the noise models before and after filtering?

**Response:** To explore the effect of noise, we used a noise-free model and 5 noise models to estimate the GPS velocities before and after filtering: white noise plus power low noise (WN+PN), white noise plus random walk noise (WN+RW), white noise plus flicker noise (WN+FN), white noise plus power low noise plus random walk noise (WN+FN+RW), and white noise plus random walk noise plus generalized Gauss-Markov (WN+RW+GGM).

(4) GNSS and GPS are inconsistency, GNSS in title but GPS throughout all the paper.
**Response:** We have changed the GNSS to GPS.

(5) P6L7-14: the discussed stations need some introductions to explain further. Are these stations with large differences to previous results?
**Response:** We have added the following introduction: "The GPS data time span also has an important effect on the velocity estimate, such as CAPF, located in NAP……" These stations present large differences relative to previous results. We think this is due to both the data and filtering (as described in Section 3.1); thus, the GPS data time span has an important effect on the velocity estimates, which are significantly different than previous results (we also compared the velocities between our results and Mart ń-Espa ñol et al., 2016, and the difference varied between 0 and 7 mm/yr), which shows that the time span will directly affect the results of the GIA evaluation. Further study is required to quantify their respective impacts, which is beyond the scope of this study.

(6) P4L17: the definition of 'residual' is same as the residual series in Section 2.1?
**Response:** It has some little difference and we have revised the "residual time series" to "the RegEM interpolated coordinate time series (the trend, annual and semiannual terms are removed)"

(7) Figure 2 and Figure 5 are not clear, difficult to see the details. It is better to expand the cales.
**Response:** We have revised the scale and provided the original vector graphs in the attachments (the name of figure 2 and figure 5 has changed to figure 3 and figure 6 )

[Figure]

[Figure]

**Figure 3.** Results of the IC1-IC8 components (the black arrows are a positive spatial response, and the red arrows are a negative spatial response)

[Figure]

**Figure 6**. GPS velocity field after applying the noise analysis and AIC filter (mm yr⁻¹)

**An assessment of GIA solutions based on high-precision GPS velocity field for Antarctica**

Wenhao Li[1], Fei Li[1,2], Shengkai Zhang[1], Jintao Lei[3], Qingchuan Zhang [1], Feng Xiao[1], Lexian Yuan[4]

[1]Chinese Antarctic Center of Surveying and Mapping, Wuhan University, Wuhan 430079, China;
10  [2]State Key Laboratory of Information Engineering in Surveying,Mapping and Remote Sensing,Wuhan University,Wuhan 430079,China;
[3]Department of Land Surveying and Geo-Informatics, Hong Kong Polytechnic University, Hung Hom 999077, Hong Kong, China;
[4]Changjiang Spatial Information Technology Engineering Co., Ltd., Wuhan 430010, China;
15  *Correspondence to*: Fei Li(fli@whu.edu.cn)

**Abstract.** Past changes in mass loading, especially LGM (Last Glacial Maximum), may cause the viscoelastic response of the Earth, this phenomenon is the so-called glacial isostatic adjustment (GIA). GIA includes the horizontal and vertical motions of the crust, the gravity field and rotation axis of the earth. Due to the uncertainties in the ice loading history and the mantle viscosity, modeling GIA will be difficult and challenging in Antarctica. The GPS velocity field provides an effective method

to constrain the GIA vertical velocity; however, to obtain the high-precision GPS velocity field, we must consider the effects of common mode error(CME)and the choice of optimal noise model (ONM). We used independent component analysis (ICA) to remove the CME recorded at 79 GPS stations in Antarctica and determined the ONM of GPS time series based on the Akaike information criterion (AIC). Then, the high-precision GPS velocity field is obtained; we used the high-precision GPS velocity field to assess the application of GIA models in Antarctica. The results show that the maximal GPS velocity variation is up to 1.2 mm yr$^{-1}$, and the mean variation is 0.2 mm yr$^{-1}$. We find systematic underestimations of all GIA model velocities in the Amundsen Sea area (ASE). Because the upper mantle viscosities in the NAP are lower than those in the south Antarctic Peninsula (SAP), the GPS vertical velocities in NAP regions are larger than SAP regions. In the Filcher-Ronne Ice Shelves(FRIS), the observed GPS velocity and predicted GIA model velocity are consistent. In East Antarctica (EA), the vertical motion is nonsignificant, and the GIA and ice loading have a small impact in this area.

**1 Introduction**

GIA is the solid Earth's viscoelastic response to past changes in ice-ocean loading. GIA influences crustal displacements, the geoid and regional sea level patterns(Wang et al., 2008;Ivins et al., 2013;Argus et al., 2010; Hao et al., 2016); we can obtain the GIA vertical velocity through forward models (Peltier, 2004), inverse models (Riva et al. 2009) and geodetic observations (such as GPS; King et al., 2010). In the forward models, the ice model and the earth model are combined to compute the GIA velocities (Velicogna and Wahr 2006; Sasgen et al., 2007;) and the GIA vertical velocities can also be obtained by inversing other geodetic method, such as satellite altimetry and gravimetry technologies (Riva et al., 2009; Gunter et al., 2014). Differences in predictions of GIA for Antarctica persist due to the uncertainties of forward models in both the deglaciation history and Earth's rheology, but without adequate and accurate deglaciation history data, Earth structure models are greatly simplified in forward models, and the constraint data are poor in inverse models; thus, large differences in GIA persist for Antarctica. The GPS can record vertical land motion(VLM) and which has been used widely to constrain GIA uplift (Argus et al., 2014a; Peltier et al., 2015) or using a data-driven approach to directly solve for GIA (Wu et al., 2010). The actual GPS velocities are usually affected by two factors: CME and the ONM,  therefore, when using the GPS velocity field to assess or extract the GIA signal, we must filter the CME and confirm the ONM. CME are thought to be related to the spatiotemporal distribution containing unmodeled signals and errors, including environmental loading effects (Atmospheric, non-tide, hydrology, etc.) and systematic errors (Dong et al., 2006). The detrimental effects of these errors could be effectively reduced after applying filtering.

Wdowinski et al. (1997) introduced Stacking to remove the CME of GPS time series in southern California. However,  in spatial scale, we cannot describe the physical mechanism and effect of CME quantitatively. Dong et al. (2006)used principal component analysis (PCA) to analyze 5-year GPS time series in southern California. Since then, many researchers used widely PCA and modified PCA to remove the CME of GPS time series (Serpelloni et al., 2013; Shen et al., 2014; He et al., 2015; Li et al., 2015). However, CME derived PCA methods are usually considered to contain colored noise (Dong et al., 2006; Yuan

et al., 2008). In addition, PCA method is based on second-order statistics and cannot take full advantage of higher-order statistics. Therefore, PCA filtering would result in contamination when applied to non-Gaussian GPS time series.

Relative to PCA, independent component analysis (ICA) can take full advantage of higher-order statistics to exploit the non-Gaussian features of the GPS time series (Hyvärinen & Oja 2000). Ming et al. (2017) adopted ICA for an investigation of 259 GPS stations in China. Li et al.(2019) compared the filtering results of Antarctica GPS residual time series derived from PCA and ICA. Considering the shortcomings of stacking and PCA filters, we apply ICA method to extract the CME of GNSS time series from Antarctica.

The noise model is another important factor which can affect the precision of velocity estimate. Previous studies shown that the GPS time series not only contain white noise (WN) but also colored noise, e.g., flicker noise (FN) and random walk noise (RW) ( Zhang J et al.,1997 ; Mao A et al.,1999; Alvaro Santamarín-Gómez et al.,2011; Bogusz J and Klos A, 2016). If we ignored the effects of colored noise, the uncertainty of GPS velocity will be overestimated by a factor of 4 or even one order of magnitude higher than the signal amplitude (Yuan et al., 2008). For Antarctica which has a vast spatial area and complex terrain, it is not sufficient to reasonably and effectively model all GNSS station time series with only one noise model. In this paper, we adopted five noise models to confirm the ONM for the GPS time series in Antarctica: white noise plus power low noise (WN+PN), white noise plus random walk noise (WN+RW), white noise plus flicker noise (WN+FN), white noise plus power low noise plus random walk noise (WN+FN+RW), and white noise plus random walk noise plus generalized Gauss-Markov (WN+RW+GGM).

After regional filtering and confirming the ONM, we obtain the high-precision GPS velocity field, and 7 GIA models are assessed by the GPS velocity field: ICE-6G (VM5a) (Argus et al.,2014;Peltier et al.,2015),ICE-5G (VM2_L90) (Peltier et al.,2004:Argus et al.,2010),WANG (Wang et al.,2008),W12a (Whitehouse et al.,2012a、2012b),Geruo13 (Geruo et al.,2013),IJ05-R2 (Ivins et al.,2013)and Paulson07(Paulson et al.,2007), The Geruo13 model has three submodels based on different truncation orders and Gauss filtering radii: (a) truncated to 100 order and no Gauss filtering; (b) truncated to 60 order and 200 km Gauss filtering; and (c) truncated to 40 order and 500 km Gauss filtering. The IJ05-R2 model has two submodels based on different parameters of the Earth model: (a) the lithosphere thickness is 65 km and the viscosity of the lower mantle is $1.5 \times 1021$ Pa.s; and (b) the lithosphere thickness is 115 km and the viscosity of the lower mantle is $4 \times 1021$ Pa.s. In this paper, we use Geruo13 (100 order) and IJR2-05 (65 km).

Martín-Español et al.(2016) used the elastic-corrected GPS vertical velocities in Antarctica over the period 2009-2014 to asses 8 GIA models, including forward and inverse methods, they found systematic underestimations of the GPS rates over specific regions characterized by low mantle viscosities and thin lithosphere. Liu et al. (2018) applied ICA and PCA for 53 GPS stations from 2010 to 2014, and used white noise plus power law (PL) noise model to estimate GPS velocities, after correction for elastic effects, they assessed the agreements of 4 GIA models and GPS vertical velocities. They found the agreements of the GPS observed velocities and GIA models are generally improved after the spatiotemporal filtering. Martín-Español et al.(2016) and Liu et al.(2018) used 53 GPS stations' velocities to assess the GIA models, but Martín-Español et al.(2016) without the filtering, and both them considered only one noise model, there is no uniform criterion that if we need to

consider the effects of CME and noise models, therefore, a quantify study of the effects is needed in GPS velocities estimate and GIA assessment. In this paper, we used more than 79 stations with long time series (around 9 years) to achieve the confident velocity, then the influences of common mode error(CME) and 5 noise models on GPS accuracy were analyzed, finally, we assessed the application of GIA models in Antarctica.

5       The remainder of this paper is organized as follows. In section 2, the data processing and methods are briefly reviewed. The results of the processed GPS data and GIA model assessment are discussed in section 3. In section 4, we discuss the assessment results of different regions. The conclusions of our findings are presented in section 5.

**2 Data processing and methods**

**2.1 GPS data**

10    The GPS time series are downloaded from the Nevada Geodetic Laboratory(NGL). GPS time series were processed by GIPSY OASIS II software at the Jet Propulsion Laboratory (JPL), and the JPL's final orbit products were applied. Precise point positioning to ionospheric-free carrier phase and pseudorange data were used. The Global Mapping Function was applied to model tropospheric refractivity, with tropospheric wet zenith delay and horizontal gradients estimated as stochastic random-walk parameters every 5 min (Bar Sever et al., 1998). Coefficients were used to compute ocean loading for the site motion

15    model, for which the FES2004 tidal model was applied, and ocean loading was also computed in the CM frame. Finally, ambiguity resolution was applied to double differences of the estimated one-way bias parameters (Blewitt, 1989) using the wide lane and phase bias (WLPB) method, which phase-connects individual stations to IGS stations in common view (Bertiger et al., 2010).The station coordinates were converted to the IGS08 frame using daily 7-parameter transformations.

      Based on the distribution and integrity of the GPS time series, we selected 79 GPS stations with a time span from 8
20    February 2010 to 23 June 2018. The average proportion of missing data of our time series is 25.54%. Figure 1 shows the locations of the 79 GNSS stations in Antarctica. We used the third quartile criterion to removed abnormal data from the raw time series, then we subtracted these trends, annual and semiannual terms to form the residual time series by hector (the offsets estimation were based on the information http://geodesy.unr.edu/NGLStationPages/steps.txt.). For the missing values, we used the regularized expectation-maximization (RegEM) (Schneider 2001) algorithm to interpolate data and obtain the completed

25    time series. We used the completed time series to performed an ICA regional filter. Then, we confirmed the ONM for all GPS time series based on AIC. Finally, we used the high-precision GPS velocity field to assess the 7 GIA models.

**2.2 ICA filter**

As presented by previous authors (Hyvärinen & Oja 2000, Ming et al, 2017), if we want to get statistically independent components (ICs) from mixed-signals, we need to maximize the non-Gaussian characteristic of the output. Each observation

30    $X_i(t) = [x_1(t), x_2(t), \cdots, x_n(t)]^T$ can be considered as a compound of the original signals $S_i(t)$ ,but the weights are different from each other. ICA method would get a separating matrix B, and then the signals $Y_i(t)$ and best estimates of $S_i(t)$. When applied

ICA to GPS time series, each row vector x in X is the GPS coordinate series with trend and mean items removed. To remove CME using ICA, we first need to whiten the GPS time series using $Z = MX$ and $E(ZZ_T) = I$ (unit matrix), where M represents the whitened matrix and Z presents the whitened variables, and then we use ICA method to obtain a rotation matrix C, and maximize the non-Gaussian character of the projection $Y = C^T Z$. In this paper, we used the FastICA algorithm (Hyv ̈arinen 1999; Hyv ̈arinen & Oja 2000) to estimate the IC Y. The detailed description of the ICA filtering can be found in Liu et al.2018 and li et al. 2019.

First, we used a parallel analysis (PA) to confirm how many ICs are statistically significant. The PA analysis is a Monte Carlo-based simulation method which compares the observed eigenvalues with those simulated datasets. If the associated eigenvalue is larger than 99% of the distribution of eigenvalues derived from random data and the IC is retained (Peres-Neto and Jackson et al.,2005). To investigate the influence of colored noise, we compared the simulation results using and without colored noise. The colored noise was generated by Fakenet (Agnew et al.2013). Figure 2 is the PA test results of ICs using and without colored noise data, from which we can see the first 7 eigenvalues are statistically significant, and colored noise has little influence, to avoid missing some information, we use the first 8 ICs to ICA filtering.

Figure 3 shows the spatial responses of IC1-IC8, from which we can conclude that IC2 has a uniform spatial coherence; IC4 and IC8 are neither completely random nor identical, but they exhibit obvious spatially uniform localized patterns or strong spatial coherence across the network; IC7 exhibits spatially uniform localized patterns in some areas, but the pattern is not entirely uniform, which we suppose is because the unmodeled signals, local effects, and other factors are not considered herein. Based on the spatial response, we used IC2, IC4, IC7, and IC8 to extract CME.

Figure 4 is the  RegEM interpolated coordinate time series (the trend, annual and semiannual terms are removed) and raw time series of GMEZ before and after applying ICA filtering. Clearly, the scattering in the filtered time series is effectively reduced by the ICA filter, as the mean root mean square (RMS) values decrease from 6.41 mm to 4.46 mm, the maximum reduction in RMS value is 48.41%, the minimum value is 10.83%, and the mean value is 30.81%.

Figure 5 shows the RMS values of the residual time series before and after applying the ICA filters. The color bar is the the RMS reduction percentage; notably, the RMS values have a larger reduction in the SAP and the FRIS; the reductions in RMS values near the coast are smaller than those in the Antarctic interior regions.

We compared the environment loading and the ICA-extracted CME at site CAS1 (Figure 5), and the results show that CME amplitudes from ICs are not consistent with environment loading (atmospheric, non-tidal ocean, and continental water loading) the environment loading  data can download from EOST Loading Service http://loading.ustrasbg.fr/displ_all.php. We checked the other sites and obtained the same results. We also computed the correlation between the CME from each IC and each loading model, and the results were poor. Furthermore, we computed the correlation between the sum of CMEs and the sum of loading displacements and obtained the same conclusion. Therefore, we think that the ICs of the CME cannot be explained by mass loadings and they are probably related to other non-geophysical errors, such as poorly modeled orbits or unmodeled tropospheric delay (Feng et al,2017).

**2.3 AIC criterion and noise analysis**

For the precision of GNSS coordinate time series, the noise model is one of the most important factors, the ONM will be quite different because of local effects among the stations in a network. It is not sufficient to reasonably and effectively model all GNSS station time series with only one noise model. We use the AIC (Akaike,1974; Schwarz,1978) to confirm the qualities of the selected noise models. The definition of the log-likelihood is as follows:

$$ln(L) = -\frac{1}{2}[Nln\,(2\pi)+lndet(C)+ \mathrm{r^T}C^{-1}] \tag{1}$$

where $N$ is the actual number of GPS observations (gaps do not contain), and r is the residual vector of the time series. The covariance matrix $C$ is decomposed as follows:

$$C = \sigma^2\bar{C}, \tag{2}$$

where $\bar{C}$ represents the sum of different noise models, and σ is the standard deviation of the conducting WN process, where σ is estimated from the residuals:

$$\sigma = \sqrt{\frac{\mathrm{r^T}\bar{\mathrm{C}}^{-1}\mathrm{r}}{N}} \tag{3}$$

Then, the $AIC$ can be defined as follows:

$$AIC = 2k + 2\,ln(L) \tag{4}$$

Because $detc\mathrm{A} = \mathrm{c^N}\,detA$, the following formulation is implemented for the likelihood:

$$ln(L) = -\frac{1}{2}[Nln(2\pi) + lndet(\bar{\mathrm{C}}) + 2Nln(\sigma) + N]. \tag{5}$$

$k$ is the sum of the parameters in the design matrix and the noise models. The minimum AIC value is the better model.

To determine the ONM for Antarctica, we use a combination of 5 noise models supplied by Hector (Bos et al., 2013) to analyze the 79 GNSS station time series based on AIC: WN+PN, WN+RW, WN+FN, WN+FN+RW, and WN+RW+GGM. The noise analysis results for the corresponding velocities listed in Table 1 show that the WN+FN ONM accounts for 22% (18 GPS stations), the WN+RW+GGM model accounts for 5.1% (4 GPS stations), and the WN+PN model accounts for 72.2% (57 GPS stations). Furthermore, we calculate the PN spectral index and find that most of the PN spectral index approximates the FN, which indicates that the PN essence is similar to that of FN in Antarctica.

**3 Results**

**3.1 GPS velocity field**

After applying AIC noise analysis and ICA filters, we obtain a high-precision GPS velocity field, and then, we compare the velocity changes with the raw GPS velocity. The result shows that the maximum difference is up to 1.2 mm yr$^{-1}$ (WWAY), the mean difference is 0.2 mm yr$^{-1}$, and 21 % (17 stations) of the velocities are greater than ±0.4 mm yr$^{-1}$. We exclude 9 stations that are inappropriate percentage statistics: FIE0, BUMS, MAW1, PECE, OHI2, STEW, VESL, MCM4, and HOOZ (processed GPS velocities are far greater than the raw velocities or the velocity directions are changed before and after applying AIC and ICA). We calculate the percentage of velocities that vary relative to raw GPS velocities, the maximum variety of processed velocities is 80.22 %(ABBZ, which has a very small velocity magnitude), and the mean variety is 11.39 %. We find that the maximum velocity variety is up to 0.9 mm yr$^{-1}$, and the mean variety is 0.6 mm yr$^{-1}$ at the remaining 9 stations. Considering the elastic and GIA magnitudes, we cannot ignore these effects.

Figure 7 is the GPS velocity field after applying noise analysis and the AIC filter to Antarctica. The overall trend is upward. INMN has a maximum uplift velocity of 32.6 mm yr$^{-1}$, a mean velocity of 3.3 mm yr$^{-1}$ (TOMO were removed because of some abnormal variations) (Martin-Espnol et al., 2016). Due to the lower upper mantle viscosity and mass loss caused by the collapse of the Larsen-B Ice Shelf (Nield et al., 2014), the north Antarctic Peninsula (NAP) mean uplift velocities (5.8 mm yr$^{-1}$) are larger than those of the SAP (3.7 mm yr$^{-1}$). The FRIS mean uplift velocities (4.7 mm yr$^{-1}$) are larger than those of the Ross Ice Shelf (ROSS, 0.74 mm yr$^{-1}$). The Amundsen Sea Embayment (ASE) has a mean uplift velocity of up to 13.0 mm yr$^{-1}$, which is the maximum amount of ice mass loss (Groh et al., 2012;Barletta et al.,2018). The most stable region is the East Antarctic (EA) coast, where the mean uplift velocity is only 0.1 mm yr$^{-1}$.

The GPS data time span also has an important effect on velocity estimation, such as CAPF, located in NAP, and the vertical velocity is estimated at 15.0 ±8.4 mm yr$^{-1}$ in Argus et al. (2014) based on approximately two years of GPS data, which is far greater than our estimated value of 4.1 ±0.3 mm yr$^{-1}$ in this study. ROB4 is located on the west coast of the Ross Ice Shelf, and the vertical velocity is estimated at 1.1 ±0.2 mm yr$^{-1}$, which is similar to the 2.2 ±3.2 mm yr$^{-1}$ estimated in Argus et al. (2014) based on approximately six years of GPS data and is dramatically different from the 7.5 ± 2.6 mm yr$^{-1}$ estimated in Thomas et al. (2011) based on 558 days of GPS data. These differences show that the GPS data time span plays an important role in the velocity estimation, and the longer the time span is, the more reliable the velocity estimation will be.

**3.2. Elastic correction**

In Antarctica, the GPS uplift velocities are dominated by the elastic deformation due to present ice mass loading and GIA. Riva et al. (2017) shown that the elastic response has a long wavelength influence in Antarctica; they used mass loss from glaciers, ice sheets, Greenland and Antarctic ice sheets in 1902 and 2014 to determine solid Earth deformation at regional and far fields. Based on the result in Riva et al. (2017), we calculated uplift velocities at 79 GPS sites. Figure 8 shows the GPS elastic velocities in Antarctica; the Antarctic Peninsula and ASE regions have larger elastic velocities and mean magnitudes

of 2.2 mm yr$^{-1}$ and 1.0 mm yr$^{-1}$, respectively. The FRIS and ROSS regions have smaller elastic velocities, while the EA has a negative elastic response. Clearly, the estimated GIA uplift rates would be significantly contaminated and in some areas dominated by neglecting the elastic response. When applying the elastic deformation correction, we consider GPS vertical velocities are mainly caused by GIA. We use the corrected GPS velocities to assess 7 GIA models: ICE-6G (VM5a), ICE-5G (VM2), WANG (CE-4G+RF3L20, β=0.4), W12a, Geruo13, IJ-05R2, and Paulson.

**3.3. GIA assessment**

To explore the application of GIA models in different regions, we divide Antarctica into 6 subregions(Sasgen et al. , 2013;Martin et al., 2016) and show these subregions in figure 9. The station information is indexed in Table S1.

Notably, the reference frame origin of the GIA model is the center of mass (CM) of the solid Earth (CE), while the GPS velocities are estimated in the ITRF2008 reference frame, whose origin is the CM of the total Earth system. Argus et al. (2014) thought the velocities between CM and CE caused by GIA are very small, but the velocities caused by the modern ice mass loss are more significant. If the ice loss in Greenland was 200 Gt yr$^{-1}$ and there is no ice loss in other areas, then the velocity is approximately 0.2 mm yr$^{-1}$, Schumacher et al. (2018) found that the effect of the frame origin transformation on the GPS uplift rates is very small (less than ±0.2 mmyr$^{-1}$).
[revised manuscript text omitted]
: ASE is undergoing a large ice mass loss, and the GIA contribution and upper mantle have significant effects on gravity-derived ice mass variation estimates and ice-sheet stability, respectively. Moreover, the viscosity under ASE is likely underestimated ($4\times10^{18}$ pascal-second) and could shorten the GIA response time scale by decades up to a century (Barletta et al.2018). The GIA signal in low mantle viscosity regions mainly reflects significant decadal-to-centennial ice load change, and most forward models do not account for such signals; therefore, the GIA signal of forward models are substantially less than that of inverse solutions. The difference between new GIA models and GPS velocity results (after elastic correction) were compared, and the results show that important differences still remain in West Antarctica, especially in ASE and NAP (Whitehouse et al. 2019).Figure 12 shows the difference between the GPS and GIA velocities at each GPS site, and the matching results are the worst in the ASE. The GPS uplift velocities along the ASE coast have larger differences, ranging from -1.8 mm yr$^{-1}$ (THUR) to 27.3 mm yr$^{-1}$ (BERP). All GIA model predicted velocities are systematically underestimated at the INMN, BERP, and BACK stations. ICE-6G has a maximum uplift velocity of approximately 7 mm yr$^{-1}$,which has an intermediate upper mantle value among most GIA models and predicts the largest present-day uplift velocity in ASE (Barletta et al.2018), The next is W12a, which has an uplift velocity of approximately 5 mm yr$^{-1}$, and the other models are within 2 mm yr$^{-1}$. From Table 4, we know that the ASE region has the maximum WRMS, and the largest discrepancy between the GPS and GIA models is greater than 20 mm yr$^{-1}$ (INMN). Removing the INMN and BERP stations, which have large uplift velocities, reduces the WRMS values to 7.0 mm yr$^{-1}$, 5.9 mm yr$^{-1}$, 5.6 mm yr$^{-1}$, 5.9 mm yr$^{-1}$, 5.8 mm yr$^{-1}$, 5.5 mm yr$^{-1}$, and 5.6 mm yr$^{-1}$.

Seismic evidence reveals there is a very low upper mantle viscosity, about $10^{18}$ Pa s in this area (An et al., 2015; Heeszel et al., 2016), that could cause a fast response to ice mass changes at a smaller scale (Martin et al., 2016). Zhang et al. (2017) also revealed that ASE is one of the regions that has experienced the most significant ice mass loss and most significant elastic vertical crustal deformation. The stations BACK, BERP and TOMO are all located in the Pine Island Bay region, and recent studies indicate that fast ice mass loss occurs in both the Pine Island Glacier and Thwaites Glacier in this region.

ROSS: King et al. (2012) showed that the GIA signal in the Ross Ice Shelf should be close to zero by examining GRACE data. The GRACE signal should be dominated by GIA and small ocean mass changes. Nield et al. (2016) predicted the uplift velocity across Siple Coast are more than 4 yr mm-1, and GIA vertical velocities are small over the Ross Ice Shelf and Siple Coast only when upper mantle viscosities are $0.5\text{-}1.0\times10^{20}$ Pa s, which is compatible with King et al. (2012), and they also showed that Late Holocene ice load changes may have a dominant influence on defining the present uplift of this region.

In our study, the GPS velocities are from -2.3 mm yr$^{-1}$ to 7.0 mm yr$^{-1}$ in the ROSS region, and the mean velocity is approximately 0.7 mm yr$^{-1}$. All GIA model predicted velocities are consistent with the GPS observed velocities. Except for the IJ05-R2 underestimation by 0.8 mm yr$^{-1}$, the other models overestimated the velocities in this region by approximately 1.0 mm yr$^{-1}$ ~ 2.2 mm yr$^{-1}$. The W12a model has the maximum WRMS and overestimates by approximately 2.3 mm yr$^{-1}$, which suggests that the ice in W12a model was too much in LGM, or the upper mantle viscosity was too large (Martin et al., 2016).

AP: The GPS vertical velocities in the Antarctic Peninsula are generally larger than the predictions of all GIA models. This study's uplift estimate of the FONP station is 11.9 mm yr$^{-1}$, while the mean GIA prediction is 2.0 mm yr$^{-1}$. One possible cause for such a difference is the crustal elastic response to the modern ice mass change. The Prince Gustav Ice Shelf and Larsen-A Ice Shelf collapsed in 1995. The neighboring Larsen-B Ice Shelf partially collapsed in 2002 and is quickly weakening and likely to completely disintegrate before the end of the decade. Ice shelves are the gatekeepers of glaciers flowing from Antarctica toward the ocean (Martin et al., 2016). Without ice shelves, the glacial ice enters the ocean faster and accelerates the pace of global sea level rise. Thomas et al. (2011) found that the uplift velocities of the stations in this region increased obviously after the collapse of the Larsen-B Ice Shelf; for example, the velocity of PALM was 0.1 mm yr$^{-1}$ before 2002 and reached 8.8 mm yr$^{-1}$ after 2002. Except for WANG overestimating the velocities by 2.3 mm yr$^{-1}$, the GIA models generally underestimated the velocities by more than 1.07 mm yr$^{-1}$~4.4 mm yr$^{-1}$ in the NAP. ICE-6G values are relatively consistent with the GPS velocities. The GPS vertical velocities in the NAP are generally larger than those in the SAP, which agrees with Wolstencroft et al. (2015), indicating a moderately low upper mantle viscosity in SAP, even though not as low as NAP. Nield et al. (2014) used a high-resolution ice elevation change dataset to compute the elastic correction in NAP, and a comparison of the GPS results and modelled uplift indicates upper mantle viscosities of between $6\times10^{17}$ and $2\times10^{18}$ in NAP (as Zhao et al. 2016). Moreover, the results show that the lithospheric thickness and upper mantle viscosity are much lower than that in the previous study. " Zhao et al. (2016) also found a higher viscosity of the Earth in the SAP than previously reported in the NAP, and the viscosity changes in north-south gradient can be an order of magnitude over 500 km.

FRIS: FRIS is near the Weddell Sea Embayment, the crustal thickness in the transition between EAs and WAs and the mantle viscosity are moderate (An et al., 2015; Heeszel et al., 2016). The mean GPS uplift velocity is 4.3 mm yr$^{-1}$, the uplift velocities are underestimated by the ICE-5G, Geruo13, and IJ05-R2 models by 3.49 mm yr$^{-1}$, 3.5 mm yr$^{-1}$ and 0.9 mm yr$^{-1}$, respectively, and overestimated by 0.4 mm yr$^{-1}$ ~1.7 mm yr$^{-1}$ by the other models. The matching results between the GPS and GIA are better overall, so we think that the uplift is mainly caused by the GIA in this region, which agrees with the findings of (Arguset al.,2014; Martin et al.,2016).

EAs: EA is characterized by higher upper mantle viscosity than West Antarctica, with exceptionally low upper mantle viscosity on the order of 1018 to 1019 Pa s beneath some regions of West Antarctica. Across EA, spatial variations in Earth rheology are currently poorly constrained (Whitehouse et al. 2019). 
[revised manuscript text omitted]

Alvaro Santamará-Gómez, Marie-Noëlle Bouin, Collilieux, X. , Guy Wöppelmann: Correlated errors in gps position time series: implications for velocity estimates. Journal of Geophysical Research: Solid Earth, 116.DOI: 10.1029/2010JB007701,2011.

Bogusz J , Klos A . On the significance of periodic signals in noise analysis of GPS station coordinates time series. GPS Solutions, 20(4):655-664. DOI: 10.1007/s10291-015-0478-9. 2016.

Argus D. F., Peltier W. R., Drummond R., Moore, A.W.: The Antarctica component of postglacial rebound model ICE-6G_C (VM5a) based on GPS positioning, exposure age dating of ice thicknesses, and relative sea level histories. Geophysical Journal International, 198(1): 537-563. DOI: 10.1093/gji/ggu140 ,2014.

Peltier, W.R., Argus, D. F., Drummond R.: Space geodesy constrains ice age terminal deglaciation: The global ICE-6G_C (VM5a) model. Journal of Geophysical Research: Solid Earth, 120(1): 450-487.DOI:10.1002/2014JB011176 ,2015.

Whitehouse, P. L., Bentley, M. J., Le Brocq, A. M.: A deglacial model for Antarctica: geological constraints and glaciological modelling as a basis for a new model of Antarctic glacial isostatic adjustment[J]. Quaternary Science Reviews, 32: 1-24.DOI:10.1016/j.quascirev.2011.11.016 ,2012a.

Whitehouse, P. L., Bentley, M. J., Milne, G. A., Matt, A.K., Thomas L.D.: A new glacial isostatic adjustment model for Antarctica: calibrated and tested using observations of relative sea-level change and present-day uplift rates. Geophysical Journal International, 190(3): 1464-1482. doi.org/10.1111/j.1365-246X.2012.05557.x,2012b.

A.G., Wahr, J., Zhong, S.: Computations of the viscoelastic response of a 3-D compressible Earth to surface loading: an application to Glacial Isostatic Adjustment in Antarctica and Canada. Geophysical Journal International, 192(2): 557-572.DOI:10.1093/gji/ggs030,2013.

Ivins, E. R., James, T. S. Wahr, J. Schrama, E.J.O. , Landerer, F.W. , Simon, K.M.: Antarctic contribution to sea level rise observed by GRACE with improved GIA correction. Journal of Geophysical Research: Solid Earth, 118(6): 3126-3141.

DOI: 10.1002/jgrb.50208,2013.

Paulson, A., Zhong, S., Wahr, J.: Inference of mantle viscosity from GRACE and relative sea level data. Geophysical Journal International, 171(2): 497-508.DOI:10.1111/j.1365-246X.2007.03556.x, 2007.

Bar-Sever, Y. E., Kroger, P. M., Borjesson, J. A.: Estimating horizontal gradients of tropospheric path delay with a single gps receiver. Journal of Geophysical Research,103(B3), 5019. DOI:10.1029/97jb03534 ,1998.

BLEWITT G. Carrier Phase Ambiguity Resolution for the Global Positioning System Applied to Geodetic Baselines up to 2000 km. Journal of Geophysical Research, 94(B8): 187-203. DOI:10.1029/jb094ib08p10187, 1989.

Bertiger, W., Desai, S. D. , Haines, B. , Harvey, N. , Moore, A. W. , Owen, S. , Weiss J.P.: Single receiver phase ambiguity resolution with gps data. Journal of Geodesy, 84(5), 327-337.DOI:10.1007/s00190-010-0371-9,2010.

Schneider, T.: Analysis of Incomplete Climate Data:Estimation of Mean Values and Covariance Matrices and Imputation of Missing Values . J. Climate, American Meteorological Society. 14, 853—871, doi: 10.1175/1520-0442(2001)014<0853:AOICDE>2.0.CO,2,2001.

Hyvärinen, A.: Fast and robust fixed-point algorithms for independent component analysis, IEEE Trans. Neural Netw. 10(3), 626–634.DOI:10.1109/72.761722,1999.

Hyvärinen, A., Karhunen, J., Oja, E.: Independent component analysis. Wiley, New York, doi:10.1007/978-1-4614-6170-8_147,2001.

Peres-Neto, P.R., Jackson, D.A. Somers K M.: How Many Principal Components? Stopping Rules for determining the Number of Non-trival Axes Revisited. Computational Statistics & Data Analysis.. 48, 974–997, doi: 10.1016/j.csda.2004.06.015,2005.

Agnew, D. C. (2013), Realistic simulations of geodetic network data: The Fakenet package,, Seismol. Res. Lett., 84, 426–432, doi:10.1785/gssrl.84.3.426.

Bos, M.S., Fernandes, R.M.S., Williams, S.D.P., Bastos, L.: Fast error analysis of continuous gnss observations with missing data. J. Geod. 87, 351–360, doi:10.1007/s00190-012-0605-0,2013.

Akaike, H. A new look at the statistical model identification. IEEE Transactions on Automatic Control, 19(6):716–723. DOI: 10.1109/TAC.1974.1100705.1974.

Schwarz, G. Estimating the Dimension of a Model. The Annals of Statistics,6(2):461–464.DOI:10.1214/aos/1176344136.1978.

Mart´ın-Espa˜nol, A., King, M.A., Zammit-Mangion, A., Andrews, S.B.,Moore, P., Bamber, J.L.: An assessment of forward and inverse GIA solutions for Antarctica, J. geophys. Res. 121(9), 6947–6965.DOI: 10.1002/2016JB013154,2016.

Nield, G. A., V. R. Barletta, A. Bordoni, M. A. King, P. L. Whitehouse, P. J. Clarke, E. Domack, T. A. Scambos, and E. Berthier.: Rapid bedrock uplift in the Antarctic Peninsula explained by viscoelastic response to recent ice unloading, Earth Planet. Sci. Lett., 397, 32–41,doi:10.1016/j.epsl. 2014.04.019, 2014.

Groh, A., H.,Ewert, M., Scheinert, M., Fritsche, A., Ru ke, A., Richter, R., Rosenau and R. Dietrich.: An investigation of glacial isostatic adjustment over the Amundsen Sea sector, West Antarctica, Global Planet. Change, 98–99, 45–53.DOI:

10.1016/j.gloplacha.2012.08.001, 2012.

Thomas,I.D., King,M.A. , Bentley,M.J. , Whitehouse,P.L. , Penna,N.T. , Williams,S.D.P. , Riva,R.E.M. , Lavallee, D.A. , Clarke, P.J., King, E.C. , Hindmarsh, R.C.A. Koivula, H.: Widespread low rates of Antarctic glacial isostatic adjustment revealed by GPS observations, Geophys. Res. Lett., 38, L22302, doi:10.1029/2011GL049277,2011.

5    Riva, R.E.M.,Frederikse, T., King, M.A., Marzeion, B. , van den Broeke, M.R.: Brief communication: the global signature of post-1900 land ice wastage on vertical land motion, The Cryosphere, 11, 1327–1332.DOI:10.5194/tc-2016-274,2017.

Sasgen, I.,H. Konrad,E. R. Ivins,M. R. van den Broeke, J. L. Bamber, Z. Martinec, V. Klemann.: Antarctic ice-mass balance 2002 to 2011: regional re-analysis of GRACE satellite gravimetry measurements with improved estimate of glacial-isostatic adjustment, The Cryosphere, 7, 1499–1512.DOI:10.5194/tcd-6-3703-2012,2013.

10   Schumacher, M., King, M. A., Rougier. J., Sha. Z., Khan. S. A., Bamber, J. L. A new global gps data set for testing and improving modelled gia uplift rates. Geophysical Journal International, 214(3), 2164-2176.DOI:10.1093/gji/ggy235.2018.

An, M.,D. A. Wiens., Y. Zhao., M. Feng., A. A. Nyblade., M. Kanao., Y. Li., A. Maggi., J. J. Lévêque .: S-velocity model and inferred Moho topography beneath the Antarctic plate from Rayleigh waves, J. Geophys. Res. Solid Earth, 120(1), 359–383, doi:10.1002/2014JB011332,2005.

15   Rignot, E.; Bamber, J. L.; Van, d. B. M. R.; Davis, C.; Li, Y.; Van, d. B. W. J.; Van, M. E. Recent antarctic mass loss from radar interferometry and regional climate modeling. Nature Geoscience, 1(2), 106-110.DOI: 10.1038/ngeo102.2008.

Vander Wal, W., P. L. Whitehouse., E. J. O. Schrama.: Effect of GIA models with 3D composite mantle viscosity on GRACE mass balance estimates for Antarctica, Earth Planet. Sci. Lett., 414, 134–143.DOI:10.1016/j.epsl.2015.01.001

20   Ivins, E. R., and T. S. James.: Antarctic glacial isostatic adjustment: A new assessment, Antarct. Sci., 17(4), 541–553.DOI:10.1017/S0954102005002968 ,2005.

Heeszel, D. S., D. A. Wiens, S. Anandakrishnan., R. C. Aster., I. W. D. Dalziel., A. D. Huerta., A. A. Nyblade., T. J. Wilson., and J. P. Winberry.: Upper mantle structure of central and West Antarctica from array analysis of Rayleigh wave phase velocities, J. Geophys. Res. Solid Earth, 121, 1758–1775, doi:10.1002/2015JB012616,2016.

25   Zhang, B. , Wang, Z. , Li, F. , An, J. , Yang, Y. , Liu, J. . (2017). Estimation of present-day glacial isostatic adjustment, ice mass change and elastic vertical crustal deformation over the antarctic ice sheet. Journal of Glaciology, 63(240), 703-715. DOI:10.1017/jog.2017.37.2017.

Wolstencroft,M.,King,M.A. , Whitehouse,P.L. , Bentley,M.J . , Nield,G.A. , King,E.C ., McMillan, ., Shepherd, A. , Barletta, V. , Bordoni, A. , Riva, R.E.M., Didova, O., Gunter, B.C.: Uplift rates from a new high-density GPS network in Palmer
30   Land indicate significant late Holocene ice loss in the southwestern Weddell Sea, Geophys. J. Int., 203(1), 737–754, doi:10.1093/gji/ggv327,2015.

Boening, C., M. Lebsock, F. Landerer, and G. Stephens.: Snowfall-driven mass change on the East Antarctic ice sheet, Geophys. Res. Lett., 39(21), 1–5, doi:10.1029/2012GL053316,2012.

**Caption of Figures**

**Figure 1. The distribution of Global Positioning System (GPS) stations in Antarctica**

**Figure 2**. The PA test results of ICs

**Figure 3. The results of IC1-IC8 components (the black arrows are a positive spatial response, the red are a negative spatial response)**

**Figure 4. the residual time series (left) and raw time series (right) of GMEZ before and after regional filter using the ICA (blue lines are the raw time series and the orange are filtered time series).**

**Figure 5. The RMS values of the residual time series before and after applying the ICA filters (%)**

**Figure 6. Contributions of IC2, IC4,IC7,IC8 (gray lines) to CME and residual time series of atmosphere (red lines), non-tide(green lines) and continental water storage(blue lines) at site CAS1.**

**Figure 7. The GPS velocity field after applying noise analysis and AIC filter**

**Figure 8. Elastic velocity of GPS in Antarctic (mm yr$^{-1}$)**

**Figure 9. The divided 6 subregionals in Antarctic. Red represent those within the North Antarctic Peninsula; yellow those within the South Antarctic Peninsula; balck those within the Amundsen Sea Embaymen; blue those within the Filchner-Ronne Ice Shelf; light green those within the Ross Ice Shelf; and brown those along coastal East Antarctica.**

**Figure 10. The uplift velocities of GIA models (mm yr$^{-1}$)**

**Figure 11. The summary statistics WM and WRMS, Median values are indicated in brackets (mm yr$^{-1}$)**

**Figure 12. The discrepancies between the modeled and the observed GIA uplift rates estimated from different solutions computed at each GPS site (Red circles indicate places where the estimated GIA rates underestimate the observed velocities from GPS; blue circles indicate the converse.)**

[Figure]

**Figure 1**. The distribution of Global Positioning System (GPS) stations in Antarctica

**Figure 2**. The PA test results of ICs (left is the results that without colored noise and right figure is the results using colored noise. Blue Line is the GPS data, black and red lines are the maximum and minimum values of simulation results)

[Figure]

[Figure]

**Figure 3.** The results of IC1-IC8 components (the black arrows are a positive spatial response, the red are a negative spatial response)

[revised manuscript text omitted]

5 **Table 3.** The WM results in North Antarctic Peninsula (NAP), South Antarctic Peninsula (SAP), Amundsen Sea Embayment (ASE), Margins of the Ross (ROSS), Filscher-Ronne Ice Shelves(FRIS), and East Antarctica (EA)a

| Model | NAP(15) | | SAP(8) | | ASE(5) | | ROSS(25) | | FRIS(5) | | EA(8) | | Antarctica(79) | |
|---|---|---|---|---|---|---|---|---|---|---|---|---|---|---|
| | WM | WM* | WM | WM* | WM | WM* | WM | WM* | WM | WM* | WM | WM* | WM | WM* |
| ICE6G_C | -1.18 | -1.49 | 2.57 | 2.42 | -4.68 | -3.78 | 1.25 | 1.16 | 1.92 | 1.38 | -0.09 | 0.49 | 0.48 | 1.02 |
| ICE5G | -4.80 | -5.21 | -1.71 | -1.14 | -9.98 | -9.04 | 1.35 | 1.58 | -2.72 | -0.34 | -1.03 | -0.57 | -1.04 | 0.34 |
| WANG | 2.28 | 1.98 | 4.22 | 5.29 | -8.72 | -7.81 | 1.89 | 1.46 | 2.62 | 2.46 | 1.76 | 2.47 | 1.78 | 2.14 |
| W12a | -1.91 | -2.27 | -1.98 | -2.22 | -6.27 | -5.42 | 2.37 | 2.06 | 2.75 | 2.11 | 0.39 | 0.93 | 0.80 | 1.61 |
| Geruo13 | -4.51 | -4.91 | -1.64 | -1.07 | -9.89 | -8.95 | 1.33 | 1.52 | -2.65 | -0.32 | -1.12 | -0.63 | -1.01 | 0.32 |
| IJ05R2 | -3.71 | -4.06 | 0.25 | 0.59 | -10.02 | -9.14 | -0.86 | -0.91 | 0.51 | 0.73 | -0.16 | 0.65 | -1.38 | -0.65 |
| Paulson | -2.04 | -2.40 | 1.65 | 2.33 | -10.06 | -9.16 | 1.07 | 0.72 | 1.45 | 1.70 | 0.05 | 1.01 | 0.10 | 0.93 |

[a] The number of sites in each area is indicated in parentheses; * represents the results of applying ICA filtering and AIC noise analysis.

**Table 4.** The WRMS results in North Antarctic Peninsula (NAP), South Antarctic Peninsula (SAP), Amundsen Sea Embayment (ASE), Margins of the Ross and Filscher-Ronne (FRIS) Ice Shelves, and East Antarctica (EA)[a]

| Model | NAP(15) | | SAP(8) | | ASE(5) | | ROSS(25) | | FRIS(5) | | EA(8) | | Antarctica(79) | |
|---|---|---|---|---|---|---|---|---|---|---|---|---|---|---|
| | WRMS | WRMS* | WRMS | WRMS* | WRMS | WRMS* | WRMS | WRMS* | WRMS | WRMS* | WRMS | WRMS* | WRMS | WRMS* |
| ICE6G_C | 3.02 | 3.31 | 2.89 | 2.96 | 11.20 | 11.83 | 1.47 | 1.35 | 2.37 | 2.22 | 0.87 | 1.10 | 3.49 | 2.29 |
| ICE5G | 5.72 | 6.14 | 2.86 | 2.97 | 14.02 | 14.22 | 1.98 | 1.92 | 3.16 | 2.52 | 1.37 | 1.29 | 4.61 | 2.95 |
| WANG | 3.62 | 3.57 | 5.32 | 6.35 | 13.28 | 13.55 | 2.55 | 1.93 | 4.09 | 3.99 | 2.94 | 3.14 | 4.70 | 3.47 |
| W12a | 3.53 | 3.88 | 2.85 | 3.46 | 11.90 | 12.37 | 2.91 | 2.67 | 3.59 | 3.01 | 0.99 | 1.49 | 4.27 | 3.37 |
| Geruo13 | 5.47 | 5.88 | 2.81 | 2.93 | 13.96 | 14.16 | 1.93 | 1.85 | 3.09 | 2.50 | 1.45 | 1.32 | 4.54 | 2.9 |
| IJ05R2 | 4.76 | 5.14 | 1.80 | 2.35 | 14.13 | 14.26 | 1.32 | 1.22 | 2.15 | 2.32 | 1.10 | 1.45 | 4.24 | 2.68 |
| Paulson | 3.58 | 3.96 | 2.88 | 3.61 | 14.14 | 14.30 | 1.64 | 1.16 | 2.58 | 2.83 | 1.40 | 1.54 | 4.22 | 2.69 |

[a] The number of sites in each area is indicated in parentheses; * represents the results of applying ICA filtering and AIC noise analysis.

---

## Author Comment (AC2) · 2 Oct 2019

The comment was uploaded in the form of a supplement:
https://www.solid-earth-discuss.net/se-2019-101/se-2019-101-AC2-supplement.pdf